# BORT: TOWARDS EXPLAINABLE NEURAL NETWORKS WITH BOUNDED ORTHOGONAL CONSTRAINT

**Borui Zhang , Wenzhao Zheng , Jie Zhou , Jiwen Lu**[*]
Department of Automation, Tsinghua University, China
Beijing National Research Center for Information Science and Technology, China
{zhang-br21, zhengwz18}@mails.tsinghua.edu.cn; {jzhou, lujiwen}@tsinghua.edu.cn

## ABSTRACT

Deep learning has revolutionized human society, yet the black-box nature of deep neural networks hinders further application to reliability-demanding industries. In the attempt to unpack them, many works observe or impact internal variables to improve the **comprehensibility** and **invertibility** of the black-box models. However, existing methods rely on intuitive assumptions and lack mathematical guarantees. To bridge this gap, we introduce **Bort**, an optimizer for improving model explainability with **B**oundedness and **ort**hogonality constraints on model parameters, derived from the sufficient conditions of model comprehensibility and invertibility. We perform reconstruction and backtracking on the model representations optimized by Bort and observe a clear improvement in model explainability. Based on Bort, we are able to synthesize explainable adversarial samples without additional parameters and training. Surprisingly, we find Bort constantly improves the classification accuracy of various architectures including ResNet and DeiT on MNIST, CIFAR-10, and ImageNet. Code: https://github.com/zbr17/Bort.

## 1 INTRODUCTION

The success of deep neural networks (DNNs) has promoted almost every artificial intelligence application. However, the black-box nature of DNNs hinders humans from understanding how they complete complex analyses. Explainable models are especially desired for reliability-demanding industries such as autonomous driving and quantitative finance. Complicated as DNNs are, they work as mapping functions to connect the input data space and the latent variable spaces (Lu et al., 2017; Zhou, 2020). Therefore, we consider explainability in both mapping directions. (Forward) **Comprehensibility:** the ability to generate an intuitive understanding of how each module transforms the inputs into the latent variables. (Backward) **Invertibility:** the ability to inverse the latent variables to the original space. We deem a model explainable if it possesses comprehensibility and invertibility simultaneously. We provide the formal descriptions of the two properties in Section 3.1.

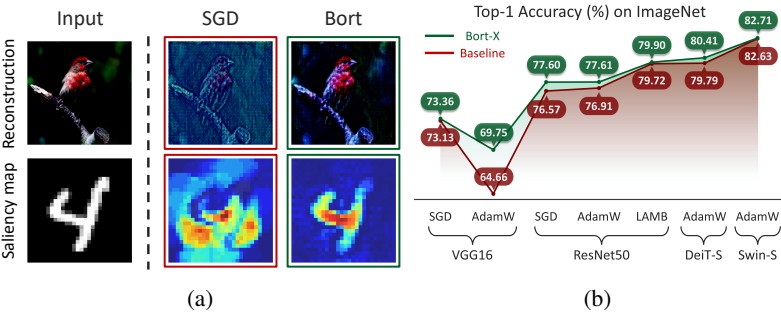

Figure 1: Bort improves explainability and performance simultaneously. (a) Examples of reconstruction and saliency analysis. (b) Top-1 accuracy with various networks and optimizers on ImageNet.

Existing literature on explainability can be mainly categorized into black-box and white-box approaches based on whether involving internal variables. Black-box explanations focus on the external behavior of the original complex model without considering the latent states (Zhou et al., 2016;

---
[*]Corresponding author.

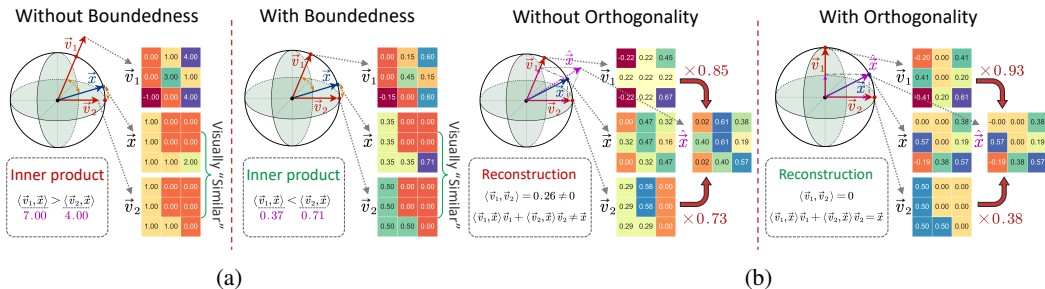

Figure 2: Motivations of the two constraints. (a) Boundedness ensures semantic similarity is consistent with dot products. (b) Orthogonality minimizes the reconstruction error for linear projection.

Lundberg & Lee, 2017; Fong & Vedaldi, 2017). For example, some methods employ simple proxy models (Ribeiro et al., 2016) to mimic the input/output behavior of the target model. They tend to produce an intuitive and coarse description of external behavior rather than an in-depth analysis of the internal mechanism of the model. In contrast, white-box explanations delve into the model to observe or intervene for a more thorough understanding. However, existing white-box explanations lack a rigorous mathematical guarantee, as shown in Figure 2. For comprehensibility, most methods (Simonyan et al., 2014; Zhou et al., 2016; Zhang et al., 2018; Liang et al., 2020) intuitively assume that the activation of feature maps is associated with the similarity between the input data and the corresponding kernel, but they provide no theoretical guarantee of the assumed "relation". For invertibility, conventional backtracking methods (Zeiler & Fergus, 2014; Springenberg et al., 2015) usually employ a linear combination of kernels layer by layer for feature reconstruction. However, they ignore the potential entanglement between kernels and thus lead to suboptimal reconstruction.

We find that almost all explainability literature is based on specific assumptions, which may be objectively incorrect or have no causal connection to the actual mechanism of the model. To bridge this gap, we give formal definitions of comprehensibility and invertibility and derive their sufficient conditions as boundedness and orthogonality, respectively. We further introduce an optimizer with **B**ounded **ort**hogonal constraint, **Bort**, as an effective and efficient instantiation of our method. Extensive experiments demonstrate the effectiveness of Bort in both model explainability and performance shown in Figure 1. We highlight our contributions as follows:

- **Mathematical interpretation of explainability**. We further derive boundedness and orthogonality as the sufficient conditions of explainability for neural networks.

- **A plug-and-play optimizer, Bort, to improve explainability.** Bort can be generally applied to any feedforward neural networks such as MLPs, CNNs, and ViTs.

- **Clear improvement of model explainability.** In addition to better reconstruction and backtracking results, we can synthesize explainable adversarial examples without training.

- **Consistent improvement of classification accuracy.** Bort improves the performance of various deep models including CNNs and ViTs on MNIST, CIFAR10, and ImageNet.

## 2 RELATED WORK

**Optimization Problem.** The properties of a trained neural network are highly affected by the optimization problem. The basic SGD optimizer updates the parameters along the stochastic gradient direction. The subsequent optimizers such as RMSProp (Tieleman et al., 2012) and Adam (Kingma & Ba, 2014) accelerate convergence by computing the adaptive gradients with the second momentum estimation and moving average. Other works focus on improving the generalization performance with a flat loss landscape (Foret et al., 2020). Additionally, the optimization constraints also affect model properties. The widely used L1 or L2 regularizations filter out redundant parameters for better generalization. AdamW (Loshchilov & Hutter, 2017) separates weight decay from the training objective to achieve this. Recent attempts adopt disentanglement constraints (Zhang et al., 2018; Shen et al., 2021; Liang et al., 2020) to improve the model explainability by forcing each filter to represent a specific data pattern. Transformation invariance constraints (Wang & Wang, 2021) later emerge to improve explainability robustness. However, these methods usually suffer from the trade-off between performance and explainability and cannot be generalized to different architectures. To break through this dilemma, we propose Bort, an optimizer with bounded orthogonal constraints, which improves both the model performance and explainability.

**Model Explainability.** The desire to understand deep neural networks promotes the development of explainable approaches over the past decade. We primarily categorize them into black-box and white-box explanations based on whether they consider the internal neural states. Black-box explanations focus on the external behaviors of a model. Saliency-based methods assign importance scores to pixels that most influence the model predictions using activation maps (Zhou et al., 2016), gradient maps (Selvaraju et al., 2017; Chattopadhay et al., 2018; Smilkov et al., 2017; Sundararajan et al., 2017; Kapishnikov et al., 2019), or perturbation maps (Petsiuk et al., 2018). Proxy-based methods approximate the input/output correlation by a simple proxy model, such as linear model (Ribeiro et al., 2016), Shapley value (Lundberg & Lee, 2017), and probabilistic model (Fong & Vedaldi, 2017; Zintgraf et al., 2017). Despite their promising results, the black-box nature prevents them from further understanding the internal mechanism of the model. Therefore, we advocate white-box methods to provide an in-depth understanding of a deep network. However, we find that existing white-box methods are usually based on ungrounded assumptions. Backtracking methods (Simonyan et al., 2014; Zeiler & Fergus, 2014; Springenberg et al., 2015) assume that each filter represents a pattern and can reconstruct input features by a weighted sum; decomposition methods (Bach et al., 2015; Shrikumar et al., 2017) believe that overall features can be expanded linearly near the reference point; hybrid-model-based methods rely on the coupled transparent rules (e.g., decision tree (Wan et al., 2020), additive model (Agarwal et al., 2021), and entropy rule (Barbiero et al., 2022)) to help understanding the internal mechanism; other methods expect disentanglement (Zhang et al., 2018; Shen et al., 2021; Liang et al., 2020; Chen et al., 2020) and invariance (Wang & Wang, 2021) constraints to regularize the parameters for better explainability. In addition, some methods (Li et al., 2018; Chen et al., 2019) try to condense the prototypes inside the model to reveal the learned concepts. We notice that only a few works (Marconato et al., 2022) try to formulate a mathematical definition of explainability, so the relationship between these assumptions and explainability lacks theoretical guarantees. To bridge this gap, we seek to define explainability mathematically for FNNs and derive its sufficient conditions to optimize an explainable network.

## 3 METHOD

In this section, we introduce the motivation and derivation of Bort in detail. Section 3.1 formulates an explainability framework including comprehensibility and invertibility properties for neural networks. Section 3.2 further derives a set of sufficient conditions (i.e., boundedness and orthogonality constraints). Finally, Section 3.3 introduces the efficient optimizer Bort and discuss its properties.

### 3.1 EXPLAINABILITY FRAMEWORK

Even though numerous efforts have explored how to define explainability descriptively (Zhang & Zhu, 2018; Gilpin et al., 2018; Bodria et al., 2021), it remains elusive to provide the mathematical definition due to its high association with the specific model type. Therefore, in this work, we concentrate on feedforward neural networks (FNN for short) and attempt to investigate the corresponding formal explainability definition. FNNs cover a large number of mainstream models, such as CNN (LeCun et al., 1995) and ViT (Dosovitskiy et al., 2020). We find that all these models can be unified under one meta-structure, a multi-layer perceptron (MLP for short) with optional nonparametric operations. For example, the convolutional layer and the transformer layer additionally use folding/unfolding and the self-attention operation, respectively. Therefore, we focus on the explainability of MLP which can be naturally generalized.

For an $l$-layer MLP $f$, we denote the dataset as $\boldsymbol{X} = \{\boldsymbol{x}_{k:1 \leq k \leq N_d} \in \mathbb{R}^{d_0}\}$ and the latent variables of each layer as $\boldsymbol{z}^i \in \mathbb{R}^{d_i}$. The overall MLP can be regarded as a composite mapping $f = f_1 \circ f_2 \circ \cdots \circ f_l$. Each layer $f_i$ is a fully-connected layer with an activation function as $\boldsymbol{z}^i = f_i(\boldsymbol{z}^{i-1}) = \sigma(\boldsymbol{W}_i \boldsymbol{z}^{i-1} + \boldsymbol{b}_i)$, where $\boldsymbol{W}_i = [\boldsymbol{w}_1^i, \cdots, \boldsymbol{w}_{d_i}^i]^T \in \mathbb{R}^{d_i \times d_{i-1}}$ and $\boldsymbol{b} \in \mathbb{R}^{d_i}$ are weight and bias parameters respectively, and $\sigma$ denotes the activation function. To understand the overall model, we start from each layer and consider both directions simultaneously.

**Forward Projection.** In this direction, information flows from input $\boldsymbol{z}^{i-1}$ to output $\boldsymbol{z}^i$. To understand the internal mechanism, we first analyze each component's functionality. It is easy to know that the activation function like ReLU (Nair & Hinton, 2010) works as the switch and the bias $\boldsymbol{b}_i$ acts as the threshold. These two components altogether filter out the unactivated neural nodes. However, we only roughly know that the weight $\boldsymbol{W}_i$ behaves like an allocator, which brings the input data to activate the most related neural node. For an explainable neural network, we argue that the row vector $\boldsymbol{w}_j^i$ in $\boldsymbol{W}_i$ should look similar to a semantic pattern, which we call **comprehensibility**. We provide the formal definition as follows:

**Definition 1** (Comprehensibility). *A weight $\boldsymbol{w}_j^i$ in FNN is said to be comprehensible if there exists a semantic pattern $\boldsymbol{z} \in \mathbb{Z}$ similar to it, which means their elements are proportional, that is*

$$\exists \boldsymbol{z} \in \mathbb{Z}, \exists k > 0, \boldsymbol{w}_j^i = k\boldsymbol{z},$$

*where $\mathbb{Z}$ represents the set of semantic data patterns.*

**Backward Reconstruction.** This direction considers how the output $\boldsymbol{z}^i$ backtracks to the input as $\hat{\boldsymbol{z}}_{i-1} = g(\boldsymbol{z}^i, f_i)$, where $g$ denotes the backtracking operation. If this backtracking operation can proceed layer by layer and ultimately reconstruct the original input data $\boldsymbol{x}$ with high precision, we call this property **invertibility**, which means that any editing of latent variables can be visually reflected by changes in the input data. The formal definition of invertibility is as follows.

**Definition 2** (Invertibility). *An FNN is said to be $\epsilon$-invertible if there exists a backtracking operation $g$ which satisfies*

$$\exists \epsilon > 0, \forall \boldsymbol{z}^{i-1}, s.t. \|\boldsymbol{z}^{i-1} - g(\boldsymbol{z}^i, f_i)\|_2 = \|\boldsymbol{z}^{i-1} - g(f_i(\boldsymbol{z}^{i-1}), f_i)\|_2 \leq \epsilon$$

### 3.2 BOUNDEDNESS AND ORTHOGONALITY

**Boundedness.** Previous explainability approaches (Zhou et al., 2016; Simonyan et al., 2014; Zeiler & Fergus, 2014; Springenberg et al., 2015; Zhang et al., 2018; Shen et al., 2021) assume that the activation value $z_j^i$ is a natural indicator, which represents the possibility that the corresponding parameter $\boldsymbol{w}_j^i$ encodes the input pattern. However, a parameter $\boldsymbol{w}_j^i$ with a high activation value is often dissimilar to the input pattern according to Definition 1. Considering $\sigma$ as a monotone function, a higher activation value indicates a larger inner product, which is computed as

$$s_j^i = \boldsymbol{w}_j^i \cdot \boldsymbol{z}^{i-1} = \|\boldsymbol{w}_j^i\| \|\boldsymbol{z}^{i-1}\| \cos\langle \boldsymbol{w}_j^i, \boldsymbol{z}^{i-1}\rangle. \tag{1}$$

This means that not only a high similarity but also a large amplitude may cause a prominent activation, as illustrated in Figure 2a. We need to ensure that if $\boldsymbol{w}_j^i$ encodes the input pattern $\boldsymbol{z}^{i-1}$, $\boldsymbol{w}_j^i$'s elements should be proportional to $\boldsymbol{z}^{i-1}$'s when training converges. To address this, we propose to restrict all $\boldsymbol{w}_j^i$ in a bounded closed hypersphere as follows:

$$\forall i, j, \|\boldsymbol{w}_j^i\|_2 \leq C_w, \text{where } C_w \text{ is a constant.} \tag{2}$$

We denote $\|\boldsymbol{z}^{i-1}\|_2$ as $C_z$, so the inner product in Eq. (1) has an upper-bound as follows:

$$s_j^i = \boldsymbol{w}_j^i \cdot \boldsymbol{z}^{i-1} \leq C_w C_z. \tag{3}$$

According to Cauchy-Schwarz inequality, $s_j^i$ takes its maximum only when there exists a non-negative $k$ such that $\boldsymbol{w}_j^i = k\boldsymbol{z}^{i-1}$, which happens to be the similarity in Definition 1. The boundedness constraint ensures that a large activation value represents a high similarity between the corresponding weight and the input pattern, which is a sufficient condition of **Comprehensibility**.

**Orthogonality.** In the FNN model, each weight $\boldsymbol{w}_j^i$ corresponds to a specific pattern. A number of approaches (Zeiler et al., 2010; Zeiler & Fergus, 2014; Springenberg et al., 2015) believe that the linear combination of these weights can reconstruct the input as follows:

$$\hat{\boldsymbol{z}}^{i-1} = g(\boldsymbol{s}^i, \boldsymbol{W}^i) = \sum_{k=1}^{d_i} \boldsymbol{w}_k^i s_k^i = \boldsymbol{W}^{i^T} \boldsymbol{s}^i, \tag{4}$$

where $s_j^i$ represents the projection of $\boldsymbol{z}^{i-1}$ onto $\boldsymbol{w}_j^i$ (i.e., inner product) and $\hat{\boldsymbol{z}}^{i-1}$ denotes the reconstructed input. We replace $g$ function in Definition 2 with Eq. (4) and formulate an optimization problem to achieve the optimal reconstruction as follows:[1]

$$\min_{\boldsymbol{W}} \mathbb{E}_{\boldsymbol{z} \sim p_{\boldsymbol{z}}} \|\boldsymbol{z} - g(\boldsymbol{W}\boldsymbol{z})\| = \mathbb{E}_{\boldsymbol{z} \sim p_{\boldsymbol{z}}} \|\boldsymbol{z} - \boldsymbol{W}^T \boldsymbol{s}\|_2^2 = \mathbb{E}_{\boldsymbol{z} \sim p_{\boldsymbol{z}}} \|\boldsymbol{z} - \boldsymbol{W}^T \boldsymbol{W} \boldsymbol{z}\|_2^2, \tag{5}$$

where $p_{\boldsymbol{z}}$ is the distribution of $\boldsymbol{z}$. We minimize Eq. (5) by letting $\nabla L = 2\mathbb{E}_{\boldsymbol{z} \sim p_{\boldsymbol{z}}}(\boldsymbol{z}\boldsymbol{z}^T)(\boldsymbol{W}^T\boldsymbol{W} - \boldsymbol{I}) = 0$, seeing Appendix A.2.1 for details. The invertibility property is expected data-independent. Thus we remove the first term $\mathbb{E}_{\boldsymbol{z} \sim p_{\boldsymbol{z}}}(\boldsymbol{z}\boldsymbol{z}^T)$ and get:

$$\boldsymbol{W}^T \boldsymbol{W} = \boldsymbol{I}, \tag{6}$$

which we call the orthogonality constraint. [2] This constraint ensures optimal reconstruction by employing Eq. (4), thus being a sufficient condition of **Invertibility**.

---

[1]We omit the superscript for brevity.

[2]To ensure Eq. (6) solvable, $\boldsymbol{W}$ requires full row rank, which means the FNN should be wide enough. Besides, the term orthogonality here means that columns of $\boldsymbol{W}^i$ should be orthogonal, not row $\boldsymbol{w}_j^i$.

### 3.3 BORT OPTIMIZER

In this section, we introduce **Bort**, an optimizer with boundedness (Eq. (2)) and orthogonality (Eq. (6)) constraints for ensuring comprehensibility and invertibility simultaneously. Let $L_t$ be the objective function. We first formulate the constrained optimization problem as follows:

$$\min_{\boldsymbol{W}^i, \boldsymbol{b}^i} L_t(\boldsymbol{X}; \boldsymbol{W}^i, \boldsymbol{b}^i, 1 \le i \le l) \tag{7}$$

$$s.t. \begin{cases} \|\boldsymbol{w}_j^i\| \le C_w, 1 \le i \le l, 1 \le j \le d_i, \\ \boldsymbol{W}^{i^T} \boldsymbol{W}^i = \boldsymbol{I}, 1 \le i \le l \end{cases}.$$

As the orthogonality constraint implies the boundedness constraint, we simplify Eq. (7) as follows:

$$\min_{\boldsymbol{W}^i, \boldsymbol{b}^i} L_t(\boldsymbol{X}; \boldsymbol{W}^i, \boldsymbol{b}^i, 1 \le i \le l) \tag{8}$$

$$s.t. \boldsymbol{W}^{i^T} \boldsymbol{W}^i = \boldsymbol{I}, \ 1 \le i \le l.$$

Then we convert Eq. (8) into an unconstrained form by utilizing the Lagrangian multiplier:

$$\min_{\boldsymbol{W}^i, \boldsymbol{b}^i} L_t(\boldsymbol{X}; \boldsymbol{W}^i, \boldsymbol{b}^i, 1 \le i \le l) + \sum_{i=1}^{l} \lambda_i \|\boldsymbol{W}^{i^T} \boldsymbol{W}^i - \boldsymbol{I}\|_F^2, \tag{9}$$

where the second term in Eq. (9) is the penalty term denoted as $L_r$, which is convex concerning $\boldsymbol{W}^{i^T} \boldsymbol{W}^i$. By calculating the derivative (derived in Appendix A.2.2), we propose **Bort** as follows:

$$(\boldsymbol{W}^i)^* \leftarrow \boldsymbol{W}^i - \alpha(\nabla L_t + \nabla L_r) = \boldsymbol{W}^i - \alpha \nabla L_t - \alpha \lambda \left(\boldsymbol{W}^i (\boldsymbol{W}^i)^T \boldsymbol{W}^i - \boldsymbol{W}^i\right), \tag{10}$$

where $\alpha$ is the learning rate and $\lambda$ is the constraint coefficient. Following Eq. (10), it is convenient to combine Bort with any other advanced gradient descent algorithm by adding an additional gradient term. Subsequently, we illustrate that the additional constraint does not limit the model capacity.

**Proposition 1.** *Given a two-layer linear model $h(\boldsymbol{x}) = \boldsymbol{v}^T \boldsymbol{W} \boldsymbol{x}$ with parameter $\boldsymbol{v} \in \mathbb{R}^m$ and $\boldsymbol{W} \in \mathbb{R}^{m \times n}$, model capacity is equivalent whether or not proposed constraints are imposed on $\boldsymbol{W}$.*

**Remark.** *We only consider the most simple case without activation functions, proved in Appendix A.2.3. Rigorous proof of keeping model capacity in general cases remains to be completed.*

Early research (Huang et al., 2006) proves that a two-layer network with random hidden nodes is a universal approximator (Hornik et al., 1989), which means that scattering latent weights benefits the property of universal approximation. Moreover, we discover in Section 4.1 that the orthogonality can even improve model performance. In addition, we design Salient Activation Tracking (SAT), a naive interpreter to take full advantage of boundedness and orthogonality (c.f. Appendix A.3).

## 4 EXPERIMENT

In this section, we evaluate the performance and explainability of Bort-optimized models. We conduct classification experiments on MNIST, CIFAR-10, and ImageNet, which shows that Bort boost the classification accuracy of various models including VGG16 (Simonyan & Zisserman, 2014), ResNet50 (He et al., 2016), DeiT (Touvron et al., 2021), and Swin (Liu et al., 2021) in Section 4.1. We also present visualization results and compute the reconstruction error to demonstrate the explainability endowed by Bort in Section 4.2. Moreover, we discover that only a few binarized latent variables are enough to represent the primary features, whereby we can synthesize the adversarial samples without additional training and parameters. Bort can be incorporated to any other optimization algorithms including SGD, AdamW (Loshchilov & Hutter, 2017), and LAMB (You et al., 2019). We denote the variant of Bort as Bort-X, where X is the first letter of the incorporated optimizer.

### 4.1 CLASSIFICATION EXPERIMENTS

#### 4.1.1 RESULTS ON MNIST/CIFAR-10

To begin with, we test Bort on MNIST (Deng, 2012) and CIFAR-10 (Krizhevsky et al., 2009). We hope to focus purely on fully-connected layers and variants (e.g., convolution layers) by eliminating potential interference (e.g., pooling layers). Therefore, we design a 5-layer all convolutional network (dubbed as ACNN-Small) by replacing all internal max-pooling layers with convolution layers with stride two (see Table 4 in the appendix for detail) following All-CNN (Springenberg et al., 2015).

Table 1: Top-1 accuracy (%) of ACNN-Small and LeNet on MNIST and CIFAR-10 datasets.

| Model | Optimizer | Setting | | | | Dataset | |
|---|---|---|---|---|---|---|---|
| | | Epoch | Lr | $\lambda_{wd}$ | $\lambda$ | MNIST | CIFAR-10 |
| LeNet | SGD | 40 | 0.01 | 0.01 | | 79.01 | 57.35 |
| | **Bort-S** | **40** | **0.01** | **0.01** | **0.1** | **88.85** (+9.84) | **62.24** (+4.89) |
| ACNN-Small | SGD | 40 | 0.01 | 0.01 | | 98.42 | 66.67 |
| | **Bort-S** | **40** | **0.01** | **0.01** | **0.1** | **99.25** (+0.83) | **72.75** (+6.08) |

**Experimental details.**   We optimize LeNet (LeCun et al., 1995) and ACNN-Small with SGD and Bort-S separately. The training recipe is quite simple. We set the learning rate to 0.01 without any learning rate adjustment schedule and train each model for 40 epochs with batch size fixed to 256. No data augmentation strategy is utilized. The constraint coefficient is set to 0.1, and the weight decay is set to 0.01. All experiments are conducted on one NVIDIA 3090 card.

**Result analysis.**   As shown in Table 1, ACNN-Small optimized by Bort-S perform significantly better than the counterpart model. We attribute this to the orthogonality constraint, which avoids redundant parameters for efficient representation. We further train a LeNet to assess the effect of other modules (e.g., pooling). We see Bort consistently boosts the classification accuracy of LeNet. This shows the internal distribution properties imposed by Bort are robust to external interference (see ablation studies in Appendix A.8).

### 4.1.2   RESULTS ON IMAGENET

We evaluate Bort on the large-scale ImageNet (Deng et al., 2009) with both CNN models (i.e., VGG16 (Deng et al., 2009) and ResNet50 (He et al., 2016)) and ViT-type models (i.e., DeiT-S (Touvron et al., 2021) and Swin-S (Liu et al., 2021)) We also combine Bort with three widely used optimizers (i.e., SGD, AdamW (Loshchilov & Hutter, 2017), and LAMB (You et al., 2019)).

Table 2: Top-1 and Top-5 accuracy (%) on ImageNet (Deng et al., 2009) dataset.

| Model | Optimizer | Epoch | Lr | BS | Top-1 | Top-5 |
|---|---|---|---|---|---|---|
| VGG16 | SGD | 300 | 0.05 | 1024 | 73.13 | 90.75 |
| | **Bort-S** | **300** | **0.05** | **1024** | **73.36** (+0.23) | **91.06** (+0.31) |
| | AdamW | 300 | 0.001 | 1024 | 64.66 | 85.11 |
| | **Bort-A** | **300** | **0.001** | **1024** | **69.75** (+5.09) | **88.72** (+3.61) |
| ResNet50 | SGD | 300 | 0.05 | 1024 | 76.57 | 92.92 |
| | **Bort-S** | **300** | **0.05** | **1024** | **77.60** (+1.03) | **93.31** (+0.39) |
| | AdamW | 300 | 0.001 | 1024 | 76.91 | 93.33 |
| | **Bort-A** | **300** | **0.001** | **1024** | **77.61** (+0.70) | **93.53** (+0.20) |
| | LAMB | 300 | 0.005 | 2048 | 79.72 | **94.53** |
| | **Bort-L** | **300** | **0.005** | **2048** | **79.90** (+0.18) | 94.37 (-0.16) |
| DeiT-S | AdamW | 300 | 0.0005 | 1024 | 79.79 | 94.72 |
| | **Bort-A** | **300** | **0.0005** | **1024** | **80.41** (+0.62) | **95.24** (+0.52) |
| Swin-S | AdamW | 300 | 0.0005 | 1024 | 82.63 | 96.02 |
| | **Bort-A** | **300** | **0.0005** | **1024** | **82.71** (+0.08) | **96.18** (+0.16) |

**Experimental details.**   In recent years, numerous approaches have improved the classification performance on ImageNet significantly. Two training recipes are involved. (1) For training CNN-type models (i.e., VGG16 and ResNet50), we follow the recipe in public codes (Wightman, 2019). We set the learning rate to 0.05 for SGD, 0.001 for AdamW, and 0.005 for LAMB. We utilize 3-split data augmentation including RandAugment (Cubuk et al., 2020) and Random Erasing. We train the model for 300 epochs with the batch size set to 1024 for SGD and AdamW and 2048 for LAMB. For LAMB, weight decay is 0.002 and $\lambda$ coefficient to 0.00002; For SGD and AdamW, we set weight decay to 0.00002 and $\lambda$ coefficient to 0.0001. (2) For ViT-type models (i.e., DeiT-S and Swin-S), we refer to the official descriptions (Touvron et al., 2021; Liu et al., 2021). We fix the batch size to 1024 and train models for 300 epochs with learning rate being 0.0005. We set weight decay to 0.005 and $\lambda$ to 0.05. Data augmentation includes RandAugment, Random Erasing, CutMix (Yun et al., 2019), and Mixup (Zhang et al., 2017). All experiments are conducted on 8 A100 cards. For more detailed training settings, we refer readers to Table 6 and Table 7 in the appendix.

**Result analysis.**   Table 2 presents the classification accuracy on ImageNet with various models and optimizers. Although Bort is an optimizer designed specifically for explainability, it is not trapped

in the trade-off between performance and explainability. The results demonstrate that Bort can significantly improve the performance of various model types, especially with SGD and AdamW. We contribute this to Bort's constraint on the parameter space, which filters out redundant parameters by orthogonality while maintaining the model capacity. In recent research, OSCN (Dai et al., 2022) has also discovered a similar phenomenon that Gram-Schmidt orthogonalization improves the performance of the conventional SCN (Wang & Li, 2017).

## 4.2 EXPLAINABILITY EXPERIMENTS

### 4.2.1 VERIFICATION OF PROPERTIES

We conduct experiments to verify the existence of orthogonality and boundedness constraints. We first train ACNN-Small models with SGD and Bort-S on MNIST separately to see whether Bort can ensure the two constraints. Then, we compute the reconstruction ratio for each layer to show the contribution of the two constraints to invertibility.

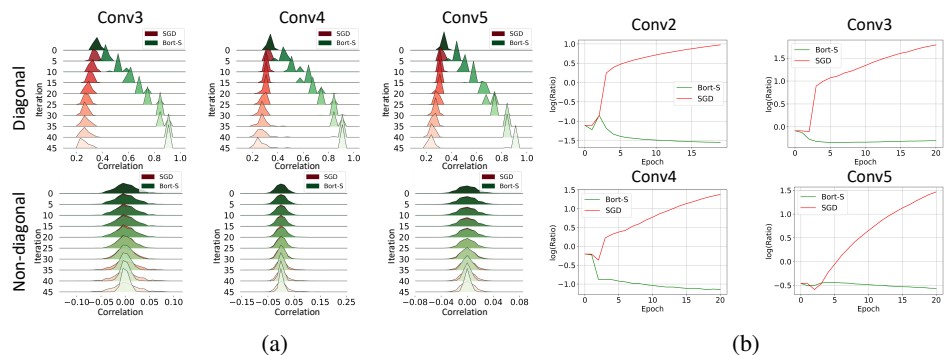

(a)                                                          (b)

Figure 3: Distribution analysis and reconstruction ratio. (a) We monitor the distribution of diagonal and non-diagonal elements of Gram Matrix. (b) We compute the reconstruction ratio of each layer.

**Distribution analysis.** Figure 3a shows the distribution of Gram Matrix $G = W^T W$, where $W$ denotes the convolution weight. We can see that Bort-S drives the diagonal elements closer to $1$ and the non-diagonal ones to $0$ while SGD with L2 regularization keeps squeezing all elements to $0$. This result demonstrates that our proposed Bort can effectively ensure the two constraints.

**Reconstruction ratio.** Following the reconstruction protocol described in Eq. (4), we compute the reconstruction error ratio as $\|z^{i-1} - \hat{z}^{i-1}\|/\|z^{i-1}\|$. Figure 3b shows that layers optimized by Bort-S can consistently reconstruct with much higher precision than SGD, demonstrating that Bort-S is significantly superior to SGD in boosting invertibility.

### 4.2.2 QUALITATIVE VISUALIZATION

In this part, we conduct reconstruction experiments and saliency analysis on MNIST, CIFAR-10, and ImageNet. Depending on the dataset size, we train the ACNN-Small (5 layers) on MNIST and CIFAR-10 and the ACNN-Base (12 layers) on ImageNet, seeing Table 4 and Table 5 for details in the appendix. We generate the visualizations using feature maps at the $5^{th}$ layer and $8^{th}$ layer of ACNN-Small and ACNN-Base, respectively.

**Reconstruction.** After training the models, we employ guided backpropagation (Springenberg et al., 2015) to reconstruct the input data (c.f. Appendix A.4). As shown in Figure 4, the model optimized by Bort-S can well preserve detailed information, such as texture and edge, during reconstruction. In contrast, the model optimized by SGD will clutter features. This phenomenon fully demonstrates that Bort can improve the invertibility of models.

**Saliency Analysis.** Exploiting boundedness and orthogonality, we design SAT algorithm to generate saliency maps (see details in Appendix A.3). Figure 5 displays the saliency map visualization results. Compared with conventional CAM (Zhou et al., 2016), our SAT approach renders more precise pattern localizations, thanks to the pixel-level feature backtracking. Moreover, the saliency maps of the model optimized by Bort concentrate more on salient objects than baseline optimizers (i.e., SGD/AdamW), proving the advantage of Bort in boosting the comprehensibility of models.

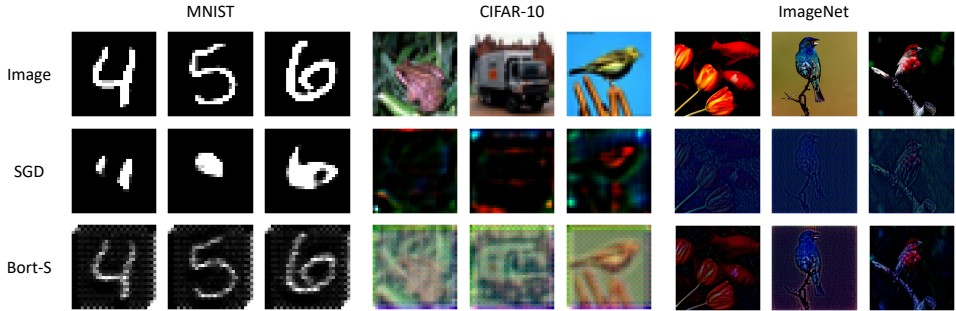

Figure 4: Reconstruction on MNIST, CIFAR-10, and ImageNet. We adopt guided backpropagation to reconstruct the input data, and our Bort achieves better reconstruction performance.

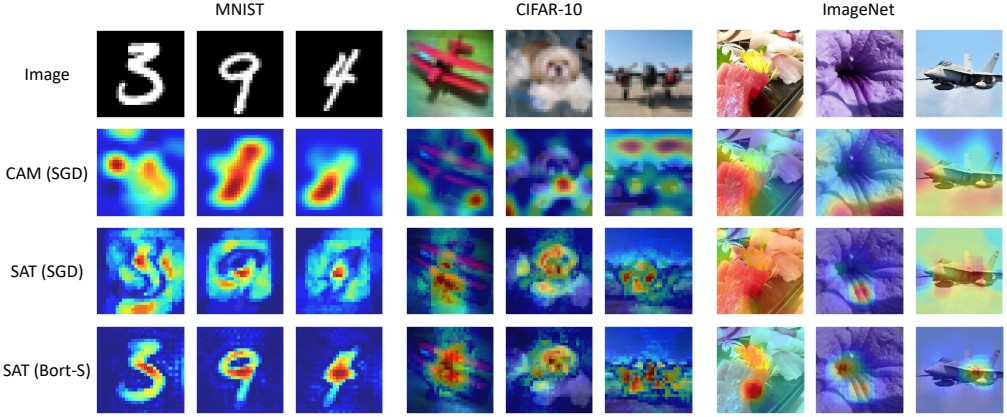

Figure 5: Generating saliency maps by CAM (Zhou et al., 2016) and our proposed SAT. We observe that SAT with Bort-S generates the best results and focuses mainly on the salient parts of objects. We set $K = 64$ for MNIST and CIFAR-10 and $K = 64$ for ImageNet.

### 4.2.3 QUANTITATIVE ANALYSIS

**Deletion/insertion metrics.** We compute the deletion/insertion metrics (Petsiuk et al., 2018) on MNIST, CIFAR-10, and ImageNet. The deletion metric measures the performance drop as removing important pixels gradually, while the insertion metric does the opposite process. For deletion, the smaller the Area Under Curve (AUC) value, the better the explainability; for insertion, a larger AUC is expected. Baselines on ImageNet and MNIST/CIFAR-10 are optimized by AdamW and SGD, respectively. As shown in Table 3, in most cases, for the common interpreters (i.e., CAM, IG, RISE, XRAI, and GuidedIG), the Deletion/Insertion metrics of models optimized by Bort are significantly better than the baseline (optimized by SGD/AdamW). Besides, we also observed that when using naive SAT, the model optimized by Bort achieved consistent improvement in all cases. We think this is because SAT takes full advantage of the boundedness and orthogonality provided by Bort.

Table 3: Insertion and deletion metrics on MNIST, CIFAR-10, and ImageNet.

| Method | Optimizer | MNIST | | CIFAR-10 | | ImageNet | |
|---|---|---|---|---|---|---|---|
| | | Deletion↓ | Insertion↑ | Deletion↓ | Insertion↑ | Deletion↓ | Insertion↑ |
| CAM | SGD/AdamW | **0.25** | **0.67** | 0.32 | 0.70 | 0.49 | 0.67 |
| | **Bort** | 0.31 (+0.07) | 0.63 (−0.05) | **0.29** (−0.04) | **0.76** (+0.06) | **0.44** (−0.05) | **0.77** (+0.10) |
| IG | SGD/AdamW | -0.04 | 0.73 | -0.37 | 0.81 | **0.07** | 0.79 |
| | **Bort** | **-0.07** (−0.03) | **0.78** (+0.05) | **-0.44** (−0.07) | **0.84** (+0.03) | 0.07 (+0.00) | **0.88** (+0.09) |
| RISE | SGD/AdamW | 0.06 | 0.64 | **0.14** | 0.75 | 0.43 | 0.75 |
| | **Bort** | **0.02** (−0.04) | **0.72** (+0.08) | 0.14 (+0.00) | **0.78** (+0.03) | **0.39** (−0.05) | **0.82** (+0.06) |
| XRAI | SGD/AdamW | **0.12** | 0.73 | 0.24 | 0.76 | 0.39 | 0.78 |
| | **Bort-S** | 0.13 (+0.01) | **0.79** (+0.06) | **0.22** (−0.02) | **0.79** (+0.03) | **0.34** (−0.04) | **0.84** (+0.06) |
| GuidedIG | SGD/AdamW | -0.04 | 0.71 | -0.28 | 0.78 | **0.06** | 0.82 |
| | **Bort** | **-0.05** (−0.01) | **0.78** (+0.06) | **-0.26** (+0.01) | **0.82** (+0.04) | 0.07 (+0.00) | **0.88** (+0.06) |
| SAT (Ours) | SGD/AdamW | 0.26 | 0.61 | 0.31 | 0.76 | 0.35 | 0.78 |
| | **Bort** | **0.05** (−0.20) | **0.80** (+0.20) | **0.27** (−0.04) | **0.81** (+0.05) | **0.32** (−0.04) | **0.84** (+0.07) |

### 4.2.4 Feature Decomposition and Adversarial Samples

In this part, we explore what Bort can provide for an in-depth understanding of networks through feature decomposition and sample synthesis (see details in Appendix A.5). We examine an extreme case where we can reconstruct the input data with partial features. Most adversarial samples rely on additional training and parameters, and only a few attempts focus on semantic adversarial sample generation (Mao et al., 2022). Therefore, we investigate whether we can achieve this without additional expense after thoroughly understanding the internal mechanism of networks.

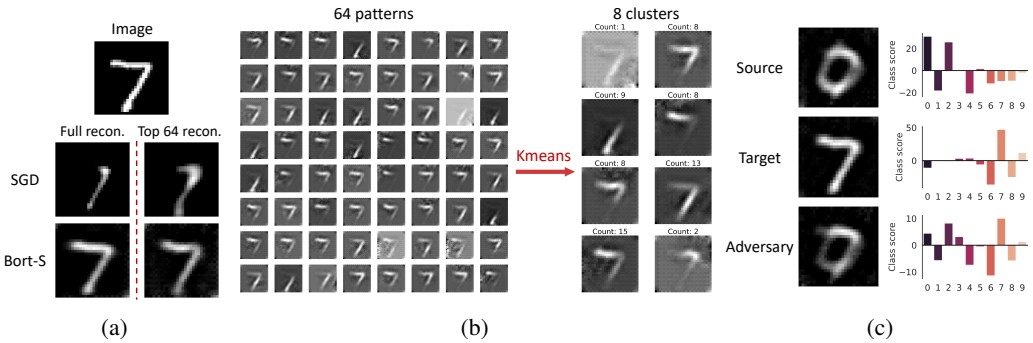

Figure 6: Feature decomposition and adversarial sample synthesis. (a) Networks optimized by Bort achieve precise reconstruction only with the 64 most salient features. (b) We visualize the top 64 features separately and run the K-Means algorithm to see their relations. (c) We synthesize semantic adversarial samples without any additional expense.

**Feature decomposition.** Given the feature map at the $5^{th}$ layer of ACNN-Small trained on MNIST, we choose the 64 most salient channels out of 2592 channels according to the maximum activations in each $6 \times 6$ feature slice. Then, we set the maximum activation to 1 at each chosen channel while setting all other activations to 0. Finally, we only keep 64 binarized variables from the original $2592 \times 6 \times 6$ variables. Figure 6a shows that even extremely sparse variables can reconstruct the input data for Bort-optimized but not SGD-optimized networks. We also reconstruct each variable and use the K-Means algorithm to cluster them as shown in Figure 6b. We observe pattern-related clusters, showing the Bort-optimized CNN is compositional and understandable.

**Adversarial sample synthesis.** For image classification tasks, most networks predict class scores using a fully-connected layer following the spatial-aggregating pooling layer. Therefore, we conjecture that spatial information is not important for classification, and we can manipulate spatial features to synthesize adversarial samples. We first choose a source and target data pair and denote their feature map as $\boldsymbol{Z}^s$ and $\boldsymbol{Z}^t$, respectively. Different from decomposition, we select the top 64 channels of source $\boldsymbol{Z}^s$ according to the target $\boldsymbol{Z}^t$ and synthesize a sparse binarized feature map $\boldsymbol{Z}^{tr}$ with them. Finally, we reconstruct $\boldsymbol{Z}^{tr}$ to obtain the adversarial sample without additional parameters and training. Interestingly, as shown in Figure 6c, the obtained adversary is semantically explainable and easily fools the classifier.

## 5 Conclusion

In this work, we provide a formal definition of explainability with comprehensibility and invertibility. We then derive two sufficient conditions (i.e., boundedness and orthogonality) and introduce the optimizer Bort to optimize FNNs efficiently with two constraints. Classification results demonstrate that by filtering out redundant parameters, Bort consistently boosts the performance of CNN-type and ViT-type models on MNIST, CIFAR-10, and ImageNet datasets. Visualization and saliency analysis qualitatively and quantitatively prove Bort's superiority in improving the explainability of networks. Surprisingly, we find that highly sparse binarized latent variables in networks optimized by Bort can characterize primary sample features, whereby we can synthesize adversarial samples without additional expense. We expect our work to inspire more research for understanding deep networks. As we derive Bort under the assumption that f is a sufficiently wide network, it would be an interesting direction to investigate the properties of Bort for narrow or extremely deep networks.

ACKNOWLEDGEMENT

This work was supported in part by the National Key Research and Development Program of China under Grant 2017YFA0700802, in part by the National Natural Science Foundation of China under Grant 62125603, and in part by a grant from the Beijing Academy of Artificial Intelligence (BAAI).

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

# A APPENDIX

## A.1 CLARIFICATION OF TERMS

**Explainability** and **interpretability** are often used interchangeably in many works of literature, although some papers actually point out subtle differences between them. In this paper, we refer to the definition in Montavon et al. (2018), where an "interpretation" maps abstract concepts into an understandable domain and an "explanation" reveals the internal mechanism (e.g., how the internal features are calculated by the model). Our Bort optimizer does not focus on the input/output behavior of the model for mapping the output features back to an understandable format (eg, image and text), but aims at revealing the internal mechanism of the black-box model by constraining the model parameters. Specifically, it includes: (1) aligning the inner product operation in FNN to the cosine similarity (comprehensibility); (2) allowing the internal features of the network to recover the features of the previous layer to the greatest extent (transparency/invertibility). We think that the property pursued by Bort is closer to the "explainability" in Montavon et al. (2018) (also similar to "model-centric" in Edwards & Veale (2017)).

## A.2 DERIVATION AND PROOF DETAILS

### A.2.1 DERIVATION OF EQ. (5)

Let $\boldsymbol{A} = \boldsymbol{W}^T \boldsymbol{W}$ and $L = \mathbb{E}_{\boldsymbol{z} \sim p_{\boldsymbol{z}}} \|\boldsymbol{z} - \boldsymbol{A}\boldsymbol{z}\|_2^2$. We compute the first-order derivate of $L$ with respect to $\boldsymbol{A}$ as follows:

$$
\begin{aligned}
\mathrm{d}L &= \mathrm{d}\mathbb{E}_{\boldsymbol{z} \sim p_{\boldsymbol{z}}} \operatorname{Tr}\left[(\boldsymbol{z} - \boldsymbol{A}\boldsymbol{z})^T (\boldsymbol{z} - \boldsymbol{A}\boldsymbol{z})\right] \\
&= \mathrm{d}\mathbb{E}_{\boldsymbol{z} \sim p_{\boldsymbol{z}}} \operatorname{Tr}\left[\boldsymbol{z}\boldsymbol{z}^T (\boldsymbol{A}^2 - 2\boldsymbol{A} + \boldsymbol{I})\right] \\
&= \operatorname{Tr}\left[2\mathbb{E}_{\boldsymbol{z} \sim p_{\boldsymbol{z}}}(\boldsymbol{z}\boldsymbol{z}^T)(\boldsymbol{A} - \boldsymbol{I})\,\mathrm{d}\boldsymbol{A}\right], \\
\nabla L &= 2\mathbb{E}_{\boldsymbol{z} \sim p_{\boldsymbol{z}}}(\boldsymbol{z}\boldsymbol{z}^T)(\boldsymbol{A} - \boldsymbol{I}).
\end{aligned}
\tag{11}
$$

To minimize $L$, we need to let the derivate be zero. Thus, we get $\nabla L = 2\mathbb{E}_{\boldsymbol{z} \sim p_{\boldsymbol{z}}}(\boldsymbol{z}\boldsymbol{z}^T)(\boldsymbol{W}^T \boldsymbol{W} - \boldsymbol{I}) = 0$.

### A.2.2 DERIVATION OF EQ. (10)

We denote the second term in Eq. (9) as $L_r = \sum_{i=1}^{l} \lambda_i \|\boldsymbol{W}^{i^T} \boldsymbol{W}^i - \boldsymbol{I}\|_F^2$. Since $\|\boldsymbol{W}^{i^T} \boldsymbol{W}^i - \boldsymbol{I}\|_2^2$ is convex with respect to $\boldsymbol{W}^{i^T} \boldsymbol{W}^i$, boundedness and orthogonality will hold at convergence if $\lambda_i$ large enough. Following the standard gradient descent algorithm, we compute the gradient of $L_r$ with respect to $\boldsymbol{W}^i$ as follows:

$$
\begin{aligned}
\mathrm{d}L_r &= \lambda_i \, \mathrm{d} \operatorname{Tr}\left[((\boldsymbol{W}^i)^T \boldsymbol{W}^i - \boldsymbol{I})^T ((\boldsymbol{W}^i)^T \boldsymbol{W}^i - \boldsymbol{I})\right] \\
&= 4\lambda_i \operatorname{Tr}\left[((\boldsymbol{W}^i)^T \boldsymbol{W}^i (\boldsymbol{W}^i)^T - (\boldsymbol{W}^i)^T)\,\mathrm{d}\boldsymbol{W}^i\right] \\
\nabla L_r &= 4\lambda_i \left(\boldsymbol{W}^i (\boldsymbol{W}^i)^T \boldsymbol{W}^i - \boldsymbol{W}^i\right).
\end{aligned}
\tag{12}
$$

For simplicity, we let $\lambda_i$ be the same, so $\nabla L_r$ becomes $4\lambda \left(\boldsymbol{W}^i (\boldsymbol{W}^i)^T \boldsymbol{W}^i - \boldsymbol{W}^i\right)$. By substitute Eq. (12) into standard gradient descent algorithm, we propose Bort as follows:

$$
(\boldsymbol{W}^i)^* \leftarrow \boldsymbol{W}^i - \alpha(\nabla L_t + \nabla L_r) = \boldsymbol{W}^i - \alpha \nabla L_t - \alpha \lambda \left(\boldsymbol{W}^i (\boldsymbol{W}^i)^T \boldsymbol{W}^i - \boldsymbol{W}^i\right),
\tag{13}
$$

where $\alpha$ is the learning rate and $\lambda$ is the constraint coefficient.

### A.2.3 PROOF OF PROPOSITION 1

*Proof.* We denote the model capacity as $\mathcal{H}_u, \mathcal{H}_c$ for unconstrained/constrained cases, respectively. (1) It is obvious that $\mathcal{H}_u \supseteq \mathcal{H}_c$ because $\mathcal{H}_c$ might be squeezed by additional constraints. (2) We then demonstrate that $\mathcal{H}_u \subseteq \mathcal{H}_c$. Given a set of configuration $(\boldsymbol{v}_0, \boldsymbol{W}_0) \in \mathcal{H}_u$, we have any data $\boldsymbol{x}$ being projected to $\boldsymbol{v}_0^T \boldsymbol{W}_0 \boldsymbol{x}$. We can decompose $\boldsymbol{W}_0$ utilizing SVD as follows:

$$
\exists \boldsymbol{U} \in \mathbb{R}^{m \times m}, \boldsymbol{V} \in \mathbb{R}^{n \times n}, \boldsymbol{\Sigma} \in \mathbb{R}^{m \times n} \; s.t. \; \boldsymbol{U}^T \boldsymbol{U} = \boldsymbol{I}, \boldsymbol{V}^T \boldsymbol{V} = \boldsymbol{I}, \boldsymbol{W}_0 = \boldsymbol{U}\boldsymbol{\Sigma}\boldsymbol{V}^T.
\tag{14}
$$

Therefore, if letting $\boldsymbol{W}_1 = \boldsymbol{V}^T$ and $\boldsymbol{v}_1 = \boldsymbol{\Sigma}^T \boldsymbol{U}^T \boldsymbol{v}_0$, we have $\boldsymbol{v}_0^T \boldsymbol{W}_0 \boldsymbol{x} = \boldsymbol{v}_1^T \boldsymbol{W}_1 \boldsymbol{x}$, which means the configuration $(\boldsymbol{v}_0, \boldsymbol{W}_0)$ and $(\boldsymbol{v}_1, \boldsymbol{W}_1)$ are equivalent. Thus $(\boldsymbol{v}_1, \boldsymbol{W}_1; \boldsymbol{W}_1^T \boldsymbol{W}_1 = \boldsymbol{I}) \in \mathcal{H}_c$. $\mathcal{H}_u \subseteq \mathcal{H}_c$ is proved. Above all, $\mathcal{H}_u = \mathcal{H}_c$. □

---

**Algorithm 1:** The SAT algorithm.

**Input:** The top feature map $\boldsymbol{Z}$, the backtracking mapping $g$, number $k$, constant $B$, and threshold $\gamma$.

**Output:** Saliency map $\boldsymbol{A}$.

1  Reset set of tuples $\mathbb{M} = \emptyset$;
2  Compute the vector $\boldsymbol{y} \in \mathbb{R}^c$ by passing $\boldsymbol{Z}$ through a max-pooling layer;
3  Get the index set of $k$ largest elements of $vy$ as $\mathbb{I}_k = \{i \mid y_i \in topk(\boldsymbol{y})\}$;
4  **foreach** $i \in \mathbb{I}_k$ **do**
5      Initiate the zero-filled $\boldsymbol{Z}^0$ with the same size of $\boldsymbol{Z}$;
6      Get the $i^{th}$ slice of $\boldsymbol{Z}^0$ as $\boldsymbol{Z}_i^0$;
7      Set the position in $\boldsymbol{Z}_i^0$ corresponding to the maximum in $\boldsymbol{Z}_i$ to constant $B$;
8      Recover the signal $\boldsymbol{S}^i$ as $\boldsymbol{S}^i = g(\boldsymbol{Z}^0)$;
9      Obtain the mask $\boldsymbol{M}^i$ by binarizing $\boldsymbol{S}^i$ through a given threshold $\gamma$;
10      Update $\mathbb{M} \leftarrow \mathbb{M} \bigcup \{(\boldsymbol{M}^i, y_i)\}$;
11  Calculate the saliency map as $\boldsymbol{A} = \sum_{(\boldsymbol{M}^i, y_i) \in \mathbb{M}} y_i \boldsymbol{M}^i$.

---

## A.3    DETAILS OF SALIENT ACTIVATION TRACKING (SAT)

**Motivation for SAT.**    We believe that mainstream interpretation methods are suboptimal for Bort because they do not take full advantage of boundedness and orthogonality, which results in Deletion/Insertion metrics not being significantly improved in a few cases, as shown in Table 3. Therefore, we germinated the idea of building a saliency map generation algorithm (SAT) for visual tasks exploiting boundedness and orthogonality.

**Implementation of SAT.**    Due to boundedness and orthogonality, the model optimized by Bort exhibits the properties of Principal Component Analysis (PCA) to some extent. Therefore, analogous to the PCA reconstruction process, SAT selects the k most salient channels of the top feature map $\boldsymbol{Z} \in \mathbb{R}^{c \times h \times w}$ for back-propagation. Note that if we do backpropagation directly, we will get features/signals instead of attribution/saliency, because saliency is more similar to masks than signals. To address this, we convert the features into masks by binarizing the reconstructed features of each channel, and calculate the final saliency map by weighted average of those masks. This design idea also appeared in RISE (Petsiuk et al., 2018). The difference is that RISE randomly samples the mask, and we calculate the mask of the k salient channel by back-propagation. We present the SAT algorithm as follows:

## A.4    DETAILS OF GUIDED-BACKPROPAGATION

We follow the standard algorithm of Guided-BP (Springenberg et al., 2015) for recovering the signals layer by layer. During the forward phase, we denote the input as $a_i$ and the ReLU layer computes the output as

$$s_i = ReLU(a_i) = \begin{cases} 0, & if\ a_i \leq 0 \\ a_i, & if\ a_i > 0 \end{cases}. \tag{15}$$

We need to store the positions where $a_i > 0$. During the back-propagation phase, given the feature $\hat{s}_i$ from the upper layer, the Guided-BP defines the backpropagation rule as

$$\hat{a}_i = GuidedBP(\hat{s}_i) = \begin{cases} \hat{s}_i, & if\ a_i > 0\ and\ \hat{s}_i > 0 \\ 0, & otherwise \end{cases}. \tag{16}$$

Other convolution layers can perform backpropagation according to Eq. (4).

## A.5    DETAILS OF DECOMPOSITION AND SYNTHESIS

Given input image $\boldsymbol{X}$, we first calculate the top feature map $\boldsymbol{Z} = f(\boldsymbol{X}) \in \mathbb{R}^{c \times h \times w}$, and get the vector $\boldsymbol{y} \in \mathbb{R}^c$ by passing $\boldsymbol{Z}$ into a max-pooling layer.

Table 4: Architecture of ACNN-Small for MNIST and CIFAR-10.

| Layer | ACNN-Small | |
| --- | --- | --- |
| | for MNIST | for CIFAR-10 |
| | Input $28 \times 28$ gray image | Input $32 \times 32$ RGB image |
| conv1 | $5 \times 5$, 8, padding 1 + ReLU | $5 \times 5$, 24 channel + ReLU |
| conv2 | $2 \times 2$, 24, stride 2 | $2 \times 2$, 64, stride 2 |
| conv3 | $4 \times 4$, 288, padding 1 + ReLU | $4 \times 4$, 512, padding 1 + ReLU |
| conv4 | $2 \times 2$, 864, stride 2 | $2 \times 2$, 1536, stride 2 |
| conv5 | $3 \times 3$, 2592, padding 1 + ReLU | $3 \times 3$, 4608, padding 1 + ReLU |
| pool | adaptive max pool | |
| softmax | 10-way softmax | |

Table 5: Architecture of ACNN-Base for ImageNet.

| Layer | ACNN-Base |
| --- | --- |
| | for ImageNet |
| | Input $224 \times 224$ RGB image |
| conv1 | $10 \times 10$, 96, stride 3, padding 4 + ReLU |
| conv2 | $1 \times 1$, 96, stride 1 + ReLU |
| conv3 | $3 \times 3$, 96, stride 2 + ReLU |
| conv4 | $3 \times 3$, 256, stride 1 + ReLU |
| conv5 | $1 \times 1$, 256, stride 1 + ReLU |
| conv6 | $3 \times 3$, 256, stride 2 + ReLU |
| conv7 | $3 \times 3$, 384, stride 1 + ReLU |
| conv8 | $1 \times 1$, 384, stride 1 + ReLU |
| conv9 | $3 \times 3$, 384, stride 2 + ReLU |
| conv10 | $3 \times 3$, 1024, stride 1 + ReLU |
| conv11 | $1 \times 1$, 1024, stride 1 + ReLU |
| conv12 | $1 \times 1$, 1000, stride 1 + ReLU |
| pool | adaptive max pool |

**Decomposition.** Analogous to PCA algorithm, we obtain the index set $\mathbb{I}$ of $k$ largest elements of $\boldsymbol{y}$. For any index $i \in sI$, we initiate a zero-filled $\boldsymbol{Z}^0$ with the same size of $\boldsymbol{Z}$. Then we set the position in slice $\boldsymbol{Z}_i^0$ corresponding to the maximum in $\boldsymbol{Z}_i$ to a constant value $B$. Finally, we perform backpropagation to get $\boldsymbol{S}^i = g(\boldsymbol{Z}^0)$. Repeating the above procedure $k$ times, we can obtain the set of recovered signals $\mathbb{S} = \{\boldsymbol{S}^i \mid i \in \mathbb{I}\}$, which is a top-k decomposition of $\boldsymbol{X}$.

**Synthsis.** Given the target feature map $\boldsymbol{Z}^t = f(\boldsymbol{X}^t)$ and the source feature map $\boldsymbol{Z}^s = f(\boldsymbol{X}^s)$, we first construct the index set $\mathbb{I}^t$ of $k$ largest elements of $\boldsymbol{y}^t$. Then we initiate a zero-filled $\boldsymbol{Z}^{tr}$. Subsequently, for each index $i \in \mathbb{I}^t$, we set the position in slice $\boldsymbol{Z}_i^{tr}$ corresponding to the maximum in $\boldsymbol{Z}_i^s$ to a constant value $y_i B$. Obviously, $\boldsymbol{Z}^{tr}$ possesses the salient channels of $\boldsymbol{Z}^t$ and the spatial information of $\boldsymbol{Z}^s$ simultaneously. Finally, we get the adversarial sample as $\boldsymbol{X}^{adv} = g(\boldsymbol{Z}^{tr})$, which may have the outlook of $\boldsymbol{X}^s$, but be classified the same as $\boldsymbol{X}^t$.

## A.6 ARCHITECTURE DETAILS

In this work, we mainly focus on fully-connected layers and the variants. Previous research (Springenberg et al., 2015) has discovered that networks only with convolution layers achieve competitive performance as conventional CNN. Therefore, we replace each internal max-pooling layer with a convolution layer (stride 2). According to the different image sizes of datasets, we design two networks (i.e., ACNN-Small and ACNN-Base) with different perceptive fields following All-CNN (Springenberg et al., 2015). The detailed architectures are displayed in Table 4 and Table 5.

Table 6: Recipes for optimization setting on ImageNet.

| Model | Optimizer | $\lambda_{wd}$ | $\lambda$ | Epoch | DropPath | Momen. | BS | LR | Sched. | Warmup |
|---|---|---|---|---|---|---|---|---|---|---|
| VGG16 | SGD | 0.00005 | | 300 | | 0.9 | 1024 | 0.05 | Cos. | 5 |
| | Bort-S | 0.00002 | 0.001 | 300 | | 0.9 | 1024 | 0.05 | Cos. | 5 |
| | AdamW | 0.00002 | | 300 | | | 1024 | 0.001 | Cos. | 5 |
| | Bort-A | 0.00002 | 0.0001 | 300 | | | 1024 | 0.001 | Cos. | 5 |
| ResNet50 | SGD | 0.00002 | | 300 | | 0.9 | 1024 | 0.05 | Cos. | 5 |
| | Bort-S | 0.00002 | 0.0001 | 300 | | 0.9 | 1024 | 0.05 | Cos. | 5 |
| | AdamW | 0.00002 | | 300 | | | 1024 | 0.001 | Cos. | 5 |
| | Bort-A | 0.00002 | 0.0001 | 300 | | | 1024 | 0.001 | Cos. | 5 |
| | LAMB | 0.02 | | 300 | | | 2048 | 0.005 | Cos. | 5 |
| | Bort-L | 0.002 | 0.00002 | 300 | | | 2048 | 0.005 | Cos. | 5 |
| DeiT-S | AdamW | 0.05 | | 300 | 0.1 | | 1024 | 0.0005 | Cos. | 5 |
| | Bort-A | 0.005 | 0.05 | 300 | 0.1 | | 1024 | 0.0005 | Cos. | 5 |
| Swin-S | AdamW | 0.05 | | 300 | 0.3 | | 1024 | 0.0005 | Cos. | 5 |
| | Bort-A | 0.005 | 0.05 | 300 | 0.3 | | 1024 | 0.0005 | Cos. | 5 |

Table 7: Recipes for loss and data setting on ImageNet.

| Model | Optimizer | AA | Mixup | CutMix | Erase | Color | AugSplit | JSD | BCD |
|---|---|---|---|---|---|---|---|---|---|
| VGG16 | SGD | m9-mstd0.5 | | | 0.6 | | 3 | ✓ | |
| | Bort-S | m9-mstd0.5 | | | 0.6 | | 3 | ✓ | |
| | AdamW | m9-mstd0.5 | | | 0.6 | | 3 | ✓ | |
| | Bort-A | m9-mstd0.5 | | | 0.6 | | 3 | ✓ | |
| ResNet50 | SGD | m9-mstd0.5 | | | 0.6 | | 3 | ✓ | |
| | Bort-S | m9-mstd0.5 | | | 0.6 | | 3 | ✓ | |
| | AdamW | m9-mstd0.5 | | | 0.6 | | 3 | ✓ | |
| | Bort-A | m9-mstd0.5 | | | 0.6 | | 3 | ✓ | |
| | LAMB | m7-mstd0.5 | 0.1 | 1 | 0 | | 3 | | ✓ |
| | Bort-L | m7-mstd0.5 | 0.1 | 1 | 0 | | 3 | | ✓ |
| DeiT-S | AdamW | m9-mstd0.5 | 0.8 | 1 | 0.25 | 0.3 | | | |
| | Bort-A | m9-mstd0.5 | 0.8 | 1 | 0.25 | 0.3 | | | |
| Swin-S | AdamW | m9-mstd0.5 | 0.8 | 1 | 0.25 | 0.4 | | | |
| | Bort-A | m9-mstd0.5 | 0.8 | 1 | 0.25 | 0.4 | | | |

## A.7 TRAINING RECIPES ON IMAGENET

Numerous attempts have explored effective techniques to boost the classification performance on ImageNet in recent years. To compare with other optimizers under fair settings, we employ two mainstream training recipes. For CNN-type networks (i.e., VGG16 and ResNet50), we follow the setting in the popular open-source library *timm* (Wightman, 2019); For ViT-type networks (i.e., DeiT-S and Swin-S), we employ the official setting described in the original papers. Detailed settings are shown in Table 6 for optimization and Table 7 for data augmentations and loss functions.

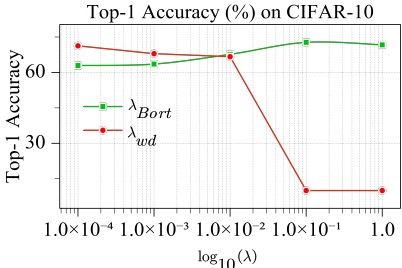

Figure 7: Ablation study about $\lambda_{wd}$ and $\lambda_{Bort}$ on CIFAR-10.

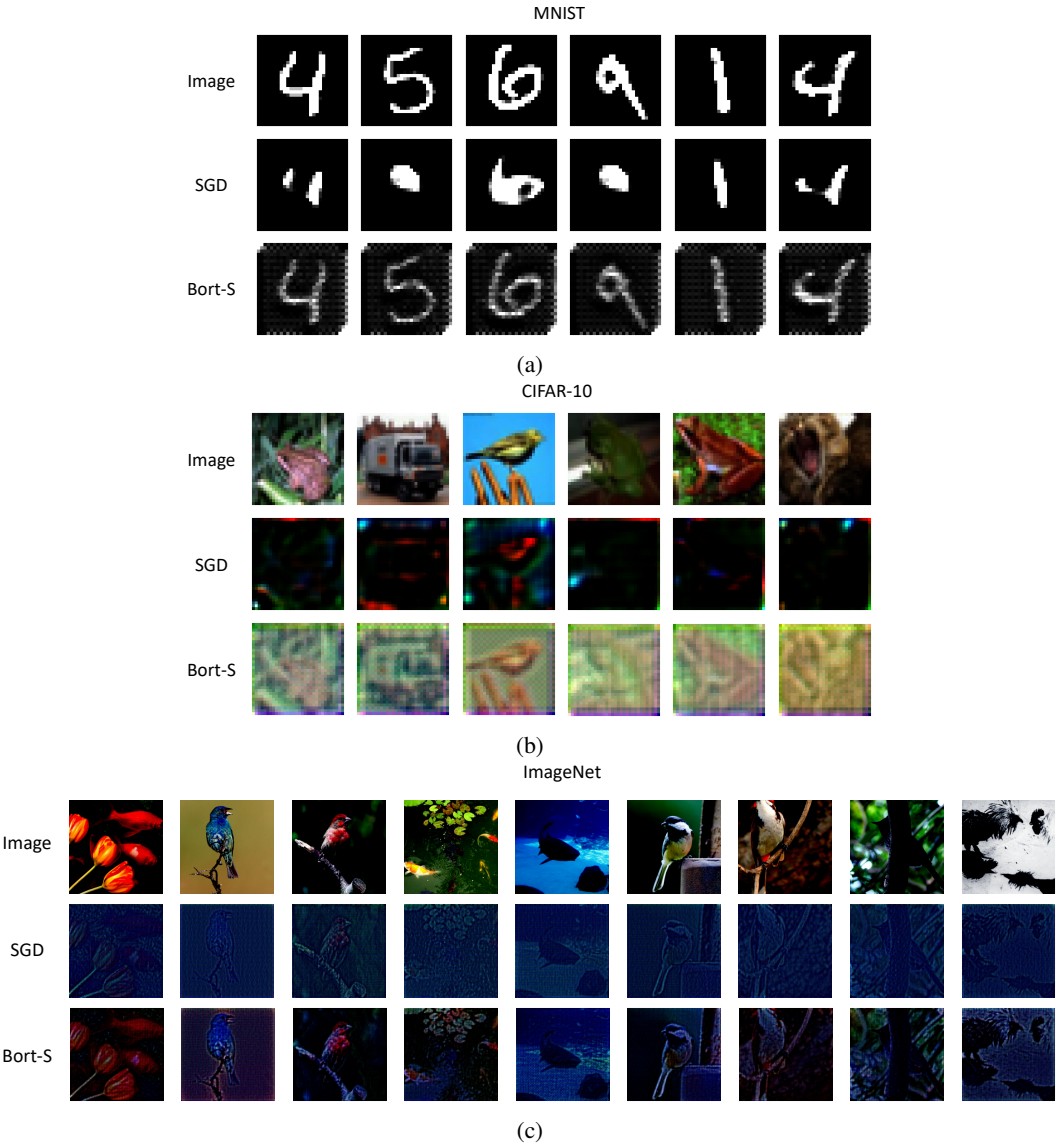

Figure 8: More reconstruction results on MNIST, CIFAR-10, and ImageNet datasets.

## A.8 ABLATION STUDY ON CIFAR-10

In this part, we explore the influence of hyper-parameters (i.e., weight decay $\lambda_{wd}$ and constraint coefficient $\lambda_{Bort}$) on CIFAR-10 dataset. Figure 7 shows that our Bort is more stable under different $\lambda_{Bort}$. In contrast, a large $\lambda_{wd}$ tends to collapse networks. We think this is because the constraints of Bort limit weights on the hyper-sphere instead of forcing them to move towards the original point.

## A.9 MORE QUALITATIVE RESULTS

We provide more visualization results in this part. To ensure the fairness of visualization, we randomly select candidates for visualization. For reconstruction results shown in Figure 8, Bort consistently boosts the reconstruction accuracy for all three datasets. Optimized by Bort, networks become invertible and easily recover most of the detailed information, such as edges and textures. For saliency maps shown in Figure 9, networks optimized by Bort better focus on the salient objects than SGD, especially for the ACNN-Small model on MNIST and CIFAR-10. We also discover that for the larger ACNN-Base model on ImageNet not all results are distinctly improved when optimized by Bort. We think this is because ACNN-Base is not wide enough to ensure perfect feature

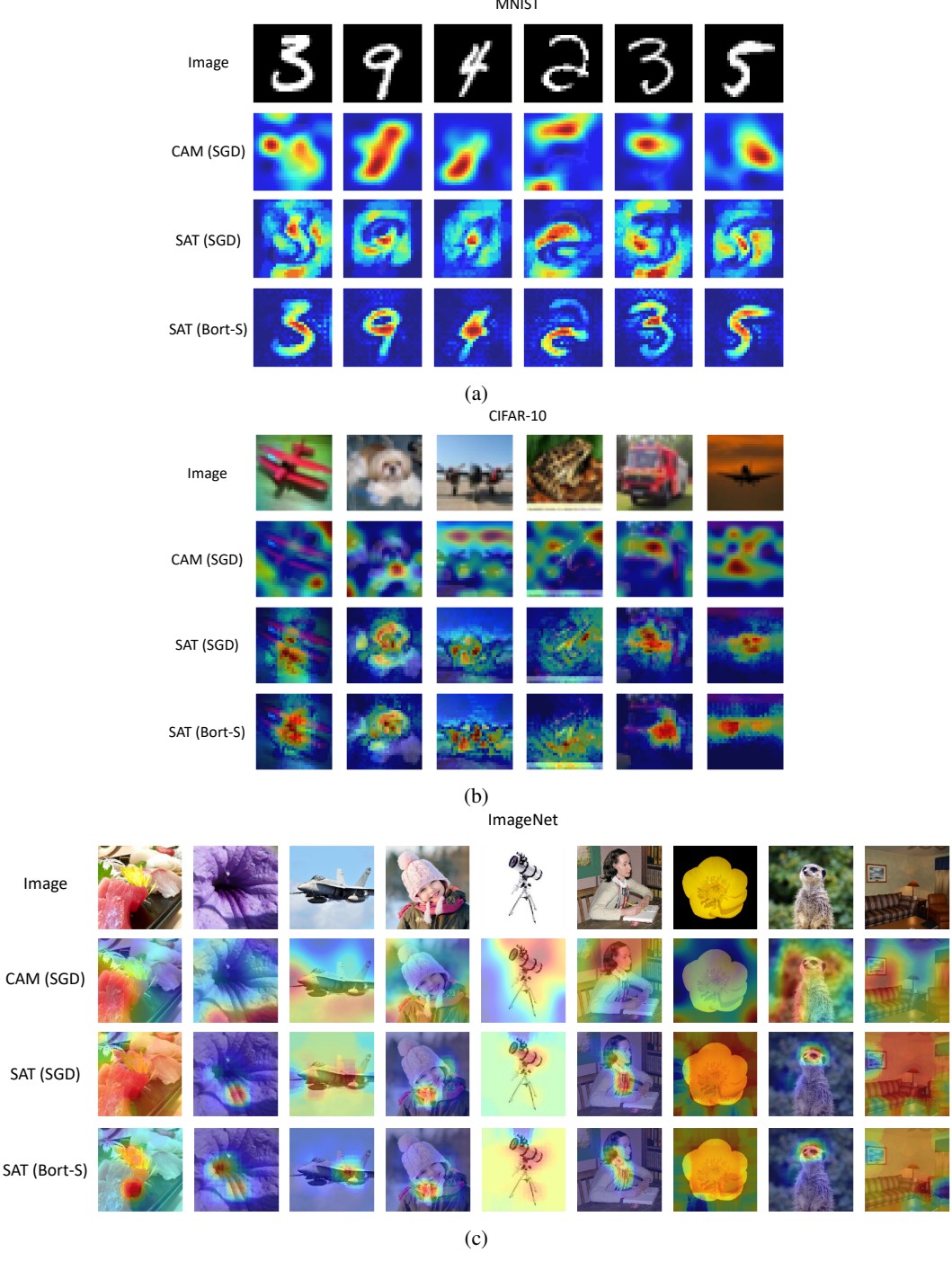

Figure 9: More saliency map results on MNIST, CIFAR-10, and ImageNet datasets.

backtracking according to the orthogonality condition (i.e., Eq. (6) is not solvable). To address this, modifying the architecture with more channels for each layer may be one possible solution, which we will investigate in the future.

