# OpenReview forum: "Bort: Towards Explainable Neural Networks with Bounded Orthogonal Constraint"
_ICLR.cc/2023/Conference — ICLR 2023 poster_

### Official Review · Reviewer_dhB8 · 2022-10-18

**Confidence:** 4
**Correctness:** 3
**Technical Novelty And Significance:** 3
**Empirical Novelty And Significance:** 2
**Recommendation:** 6

**Clarity, Quality, Novelty And Reproducibility:**

Quality

This paper is technically solid. However, it is over-ambitious as over-simplifying explainability to two measures: comprehensibility and transparency, whose definitions are debatable.

Experiments are not adequate as it only compares with CAM, a 2016 paper.

Clarity

The paper is written well in general. Please fix notations as pointed out above.

Originality

The attempt of mathematical definition and analysis of explainability is novel and plausible.

**Strength And Weaknesses:**

Strengths
+ Explainability is a critical and challenging field in ML, and the mathematical attempt proposed in this paper is encouraging.

Weaknesses
- Boundedness constraint seems similar to cosine similarity. Any difference?
- The discussion to treat CNN and Transformer as MLP is quite sketchy, e.g. "can be unified under one meta-structure, a multi-layer perceptron (MLP for short) with optional nonparametric operations. For example, the convolutional layer and the transformer layer additionally use folding/unfolding and the self-attention operation, respectively". Will parameter sharing and attention introduce extra complexity and inconsistency to the definition and analysis of comprehensibility and transparency?
-"the row vector wij in Wi". wij is a weight, right? How can a weight "look similar to a specific data pattern"? What does "look similar" mean? How can weights look similar to an input data pattern after many layers of non-linear transformation?
- The definition of comprehensibility is confusing. Is a semantic data pattern z a vector or scalar value? How will z be proportional to a weight wij? especially in later layers?
- The definitions of comprehensibility and transparency are debatable, and it is also a question whether comprehensibility and transparency are sufficient for complete explainability.
- Both boundedness and orthogonality constraints seem to limit the expressivibility of FNNs, any insight why they actually improve accuracy? some regularization effects?
- Although  a two-layer network is a universal approximator, proposition 1 is on  a two-layer linear network.

**Summary Of The Paper:**

This paper presents a new optimizer called Bort to improve model explainability (comprehensibility and transparency) with Boundedness and orthogonality constraints on model parameters.

**Summary Of The Review:**

The approach of mathematical definition and analysis of explainability proposed in this paper is novel and plausible. However, the paper is over-ambitious and experiment is not adequate.

---

> ### Author Response · Authors · 2022-11-18
> **Thanks for reviewer dhB8's suggestions and respond to relevant questions**
>
> Thanks to the affirmation and suggestion of reviewer dhB8. We have made targeted revisions and supplements. Note that we have changed the term "transparency" to "reversibility" following reviewer t9ra's comment. Here, we still use the term "transparency" for semantic coherence.
>
> 1. **Relationship between boundedness and cosine similarity**
>     - The boundedness constraint is to convert the inner product operation in FNNs to cosine similarity. According to Definition 1, the similarity is defined as the components are proportional, which is satisfied when the cosine similarity takes the maximum value.
> 2. **meta-structure of FNNs**
>     - Our work mainly constrains the boundedness and orthogonality of weights, thereby improving the comprehensibility and transparency/invertibility of parameterized modules. Non-parametric modules (e.g. parameter sharing and attention) do not affect the above two properties. We also acknowledge that our meta-structure is limited to FNNs and is not applicable to more complex structures such as models containing feedback connections.
> 3. **Definition of similarity**
>     - Sorry for the confusion. We give the definition of "look similar" in Definition 1 (i.e., the components are proportional) and the "input data pattern" refers to the input $z$ of the current module, not the original input data $x$ (we have modified definition 1 for clarity).
>     - Since conventional explainability works assume that neuron activations represent the correlation of inputs with corresponding weights. However, this correlation is calculated by the inner product, different from common similarities (e.g., cosine similarity). So we use bounded constraints to convert the inner product operation into a cosine similarity.
> 4. **Explainability**
>     - Explainability requires multi-dimensional descriptions and cannot be measured simply by a certain indicator. We mainly start from both directions of feature propagation (forward and backward) and propose comprehensibility and transparency/invertibility, which are also mentioned in prototypes ([1-2]) and back-propagations ([3-6]).
> 5. **Analysis of performance improvement**
>     - Intuitively speaking, the performance improvement of Bort comes from the maximum rank of weights, which squeezes redundant information. We give a simple two-layer linear network in Proposition 1 for illustration, and the research on stochastic configuration networks([7-8]) further demonstrates the important role of orthogonality in improving the performance.
>     - We point out the limitation of Proposition 1 in the Remark section. A complete and rigorous proof of why orthogonality facilites FNNs' performance remains a difficult problem. While this paper is not a theoretical analysis for this, we believe that exploring this issue in the future will be an important direction.
> 6. **Experimental comparison**
>     - Thanks to the reviewer for the suggestion. We have added more interpretation methods in Table 3, including Integrated Gradients (2017), RISE (2018), XRAI (2019), and GuidedIG (2021). The logic of the experimental comparison is that given the interpreter, we compare the deletion/insertion metrics of models optimized by Bort and conventional optimizers (e.g., SGD and AdamW) to verify the effectiveness of Bort on explainability. We can observe a consistent improvement for different interpreters.
>     - Note that the core goal of this paper is to establish a method to enhance the inherent explainability of models (i.e., the Bort optimizer with boundedness and orthogonality). The visualization method SAT for generating saliency maps is a by-product of exploiting these two properties.
>
> [1] Deep learning for case-based reasoning through prototypes: A neural network that explains its predictions
> [2] This looks like that: deep learning for interpretable image recognition
> [3] On pixel-wise explanations for non-linear classifier decisions by layer-wise relevance propagation
> [4] Visualizing and understanding convolutional networks
> [5] Learning important features through propagating activation differences
> [6] Striving for simplicity: The all convolutional net
> [7] Stochastic configuration networks: Fundamentals and algorithms
> [8] Orthogonal Stochastic Configuration Networks with Adaptive Construction Parameter for Data Analytics

---

> ### Author Response · Authors · 2022-11-22
> **Thanks to the reviewer for the helpful review during discussion phase 1**
>
> We thank the reviewer again for the helpful and detailed review.
>
> Since the Phase 1 discussion is over, we just want to ask if the problems raised by the reviewer are properly addressed and if we need to clarify anything else.

---

### Official Review · Reviewer_t9ra · 2022-10-24

**Confidence:** 4
**Correctness:** 3
**Technical Novelty And Significance:** 3
**Empirical Novelty And Significance:** 3
**Recommendation:** 6

**Clarity, Quality, Novelty And Reproducibility:**

**Clarity**:  The text is generally well written and well structured.  English is good, with only few grammatical issues here and there.  Pictures are also quite clear.

The formalization is relatively clean and convincing.

I think that some key terminological choices could be improved.  The terms "explainability", "comprehensibility", and "transparency" have a far more general (and fuzzily defined) meaning - in English - than the mathematical properties defined and used here.  Specifically:

 - "Explainability" has as much to do with algorithmic aspects of the explanation process as it has to do with human-centric, psychological factors such as knowledge and cognitive biases.  As such, attempting to solve it formally is (currently!) not possible.  The authors should tone down some of their claims regarding the scope of their formalization (e.g., my honest opinion is that being able to map latent features back to input space is not quite enough:  a latent point might map back to an example that has very little semantic meaning for human observer, say, a garbled image resulting from a measurement mistake or an uninformative region of a valid image.  The BORT formalization does little - and *can* do little - to prevent this from happening, see also below.)

 - What the authors called "comprehensibility" is essentially the idea that weights should recover concrete examples, i.e., prototypes.  "Prototype-based" is a better and more well-known term compared to "comprehensibility".

 - What the authors called "transparency" means that latent variables can be mapped back to input space.  This sounds more like an approximate form of "invertibility" to me.

What is confusing is that all three terms (explainability, comprehensibility, transparency) are used with different meanings in the rest of the literature.  I do encourage the authors to rename them to something more unique and easily recognizable.

Another element I did not understand is why BORT is being marketed as an optimizer.  To me it really looks like a regularization term.  Not a huge deal, just slightly confusing.


**Quality**:  The idea underlying BORT is to turn a neural network into a stack of orthogonal projections, with individual weight vectors ending up resembling a set of diverse (i.e., orthogonal) examples.  The intuition is quite clear and reasonable, and it is very much in line with prototype-based models (see below).  BORT is overall quite straightforward, but this to me is actually a plus.

One aspect I would mention in the paper is that stacks of orthogonal (linear) projections are incapable of learning, because they do preserve information whereas generalization is by its own nature lossy.  BORT works regardless because, thankfully, architectural limits (e.g., non-square weight matrices) prevent BORT-optimized neural networks from acquiring perfect orthogonal projections.

I found the empirical evaluation not entirely convincing, for several reasons:

- The experiments in Sec. 4.1 and 4.2.1 are quite clear and are positive.

- Those in Sec. 4.2.2 are very compressed and hard to parse.  It is not clear to me how guided backpropagation (a saliency technique) is used to "reconstruct the input" (Fig 4) and what the output is meant to represent: an input or a saliency map?

- Another issue with that guided backpropagation is known to be insensitive to the weights learned by the model, see:

  Adebayo J, Gilmer J, Muelly M, Goodfellow I, Hardt M, Kim B. Sanity checks for saliency maps. NeurIPS 2018.

How does this impact Fig 4?

- The **worst** issue in the paper is the description of the SAT algorithm, which is very compressed and hidden in the experiments.  SAT, despite not having been mentioned at any point in the paper, turns out to be **integral** to BORT.  In fact, using more well-known saliency techniques with BORT-trained model is not ideal (see Table 3).  Alas, I could hardly understand what SAT does.  For instance, why the result needs to be averaged over multiple images:  the way it is described, SAT sounds like a deterministic procedure so I don't see the need for averaging.  I am very confused as to why the "backprojection" SAT procedure should result in a saliency map, and what the semantics of this map are meant to be, i.e., why it can be used as an explanation.  This makes it very hard to evaluate the saliency map experiments, which are the core of the paper.

- The decomposition and synthesis experiments are also overly compressed and hard to follow.

I think SAT should be introduce much earlier in the paper, as early as the abstract, and the reason why it is needed should be clarified.  Right now it is not clear why BORT - whose main aim is to improve interpretability -  does *not* really improve interpretability unless used together with SAT.  It is very hard to evaluate BORT in isolation.


**Novelty**:  The idea of learning orthogonal transformation is not new in ML, but to the best of my knowledge this it has not been used in the context of learning deep feed-forward neural nets for the purpose of explainability.

The authors do a good job at discussing related work from the white- and black-box models, but they completely neglect the recent literature on "gray-box" (or "concept-based") models, including:

  Prototype Classification Networks (PCNs):  Li, Oscar, Hao Liu, Chaofan Chen, and Cynthia Rudin. "Deep learning for case-based reasoning through prototypes: A neural network that explains its predictions." AAAI 2018.

  Part-prototype Networks (PPNets):  Chen C, Li O, Tao D, Barnett A, Rudin C, Su JK. This looks like that: deep learning for interpretable image recognition. NeurIPS 2019.

  Concept Whitening (CW):  Chen, Z., Bei, Y. and Rudin, C.. Concept whitening for interpretable image recognition. Nature Machine Intelligence, 2020.

  GlanceNets:  Marconato E, Passerini A, Teso S. GlanceNets: Interpretabile, Leak-proof Concept-based Models. arXiv 2022.

Most of these works are somehow related to BORT:  (i) PCNs make use of autoencoders to map latent variables to input space, adding extra constraints to ensure that the reconstruction resembles a concrete example (in a vein reminiscent of BORT), (ii) CW anticipates the idea of leveraging orthogonality to enhance interpretability, although from a slightly different angle than BORT,  (iii) PCNs and PPNets work by learning prototypes, which seems very much the objective of the "comprehensibility" criterion introduced here;  (iv) GlanceNets do come with a sound formal definition of explainability, as does the work by Hazan et al (not a gray-box model, though):

  Wolf L, Galanti T, Hazan T. A formal approach to explainability. AIES 2019.

There are several other attempts at formalizing explainability out there.

I think that the authors should at the bare minimum mention the existence of gray-box models (especially those based on prototypes and orthogonalization, which are *very* close in spirit to BORT) and their relationship to the BORT desiderata, as well as highlight that there *are* alternative attempts at formalizing explainability.


**Reproducibility**:  The code is provided in the supplement.

**Strength And Weaknesses:**

PROS
- Proposed idea is is based on a solid intuition.
- Formalization is clean and looks correct.
- No performance drop upon applying BORT - actually, quite the contrary.
- Text is mostly well written.

CONS
- Key element (SAT) is not well described, complicating evaluation of the results.
- Other aspects of the experiments are not well explained.
- Some terminological choices are a bit confusing.
- Relationship to existing approaches is not entirely clear.
- A subset of the results rely on guided backpropagation, which is known to be flawed.
- Only two smaller image classification data sets (not a huge deal).

**Summary Of The Paper:**

The paper introduces two methodological components: (1) BORT, a regularization for feed-forward neural networks term that encourages the various layers to implement a bounded orthogonal projection, (2) SAT, a saliency algorithm specifically designed for BORT-regularized networks.  The empirical evaluation highlights how BORT can improve accuracy of several models on MNIST and CIFAR10, and that BORT+SAT produces "clear" saliency maps with good deletion and insertion scores (which measure sensitivity of model predictions to pixels identified as relevant by the saliency maps).  In addition, the BORT+SAT combo facilitates acquiring features with input reconstruction capabilities and generation of adversarial examples.

**Summary Of The Review:**

Explainability-enhancing regularizer based on per-layer orthonormalization + ad-hoc saliency map algorithm.  Alas, the latter is poorly described, complicating evaluation.

---

> ### Author Response · Authors · 2022-11-18
> **Thanks for reviewer t9ra's suggestions and respond to relevant questions**
>
> Thanks to reviewer t9ra for suggestion, which is of great significance to our work.
> We have made revisions and supplements, and the responses to the questions are as follows:
>
> 1. **Missing details about SAT**
>     - Sorry for the confusion. Note that our core contribution is to build a method for inherent explainability of models (i.e., the Bort optimizer). SAT for saliency maps is a by-product through exploiting boundedness and orthogonality.
>     - We have added more interpretation methods in Table 3, including IG, RISE, XRAI, and GuidedIG. To verify the effectiveness of Bort on explainability, we compare the deletion/insertion metrics of models optimized by Bort and SGD/AdamW while fixing the interpretation algorithm. Table 3 shows Bort can consistently improve deletion/insertion metrics in most cases. Therefore, Bort is not strictly bound to SAT for explainability.
>     - Due to space limitations, we have added the design motivation and the detailed formulation of SAT in Appendix A.3.
> 2. **Details about decomposition and synthesis experiments**
>     - Sorry for the confusion caused by the compressed description. The motivation for decomposition and synthesis experiments is similar to SAT, which is a reasonable use of boundedness and orthogonality. We have included details in Appendix A.5.
> 3. **Modification of some terms**
>     - **For "explainability"：** In this paper, we refer to the definition in [2], where an **interpretation** maps abstract concepts into an understandable domain and an **explanation** reveals the internal mechanism (e.g., how the internal features are calculated by the model). Our Bort optimizer does not focus on the input/output behavior of the model for mapping the output features back to an understandable format, but aims at revealing the internal mechanism of models by constraining parameters. We think that the property pursued by Bort is closer to "explainability"  in [2] (also similar to "model-centric" in [3]). So we want to continue using the term "explainability".
>     - **For "transparency":**  Thanks to the reviewer for the suggestion, we agree that invertibility is a more accurate term than transparency, and we have made corresponding modifications.
>     - **For "comprehensibility":** We agree that "comprehensibility" is similar to "prototype". However, the term we hope to use is a functional description rather than a technical one. Therefore, we still hope to continue using "comprehensibility".
> 4. **Why Bort is an optimizer**
>     - We believe that the optimization problem includes both the optimization objective and constraints, both of which affect the optimization trajectory. Some previous optimizer works focused on constraints/regularizations, such as AdamW ([4]), and Bort aims at the optimization problem with bounded orthogonal constraints. So we call Bort a new optimizer.
> 5. **Supplement to the literature review**
>     - We have supplemented the references.
> 6. **Use of guided-backpropagation**
>     - Sorry for the confusion on BP usage. We have added details on Sec. 4.2.2 in Appendix A.4.
>     - We don't think [5] conflicts with our results:
>       1. Differences in implementation details. Tested models in [5] are not "wide" networks used in our paper.
>       2. [6] also points out the problems in [5] and both vanilla BP and guided-BP pass the modified sanity checks in [6]. The conclusion in [5] is not yet a consensus.
> 7. **Deletion/Insertion metrics on ImageNet**
>     - We have added deletion/insertion results for large datasets in the modified version, as shown in Table 3.
> 8. **Theoretical analysis**
>     - Indeed, due to structure constraints (eg, the network is not wide enough), perfect orthographic projection may not be achieved, which we have stated in footnotes. However, even if the perfect orthogonal projection cannot be obtained, the Bort optimizer can achieve the maximum rank of the weights, squeezing redundant information.
>     - As for why Bort can improve the performance of the model, we give a simple two-layer linear network in Proposition 1 to illustrate, and the research on stochastic configuration networks([7-8]) demonstrates the important role of orthogonality in improving the performance. A complete and rigorous proof of why orthogonality improves models remains a difficult problem, and we believe that exploring this issue in the future will be an important research direction.
>
> [1] Rise: Randomized input sampling for explanation of black-box models
> [2] Methods for interpreting and understanding deep neural networks
> [3] Slave to the algorithm: Why a right to an explanation is probably not the remedy you are looking for
> [4] Decoupled weight decay regularization
> [5] Sanity checks for saliency maps
> [6] Revisiting Sanity Checks for Saliency Maps
> [7] Stochastic configuration networks: Fundamentals and algorithms
> [8] Orthogonal Stochastic Configuration Networks with Adaptive Construction Parameter for Data Analytics

---

> > ### Comment · Reviewer_t9ra · 2022-11-19
> > **Reply**
> >
> > Thank you for the detailed rebuttal.  Here are some comments.
> >
> > **Terminology.** I completely agree with you that the terms "explainability", "interpretability" etc. are overloaded.  This is precisely why I suggested to use less overloaded alternatives.  I appreciate that the second property is now called "invertibility".
> >
> > I wholeheartedly agree that "explainability is a multi-dimensional problem, which is difficult to completely solve with a single indicator."  The way the paper came across, however, sounded like you were proposing that orthogonality and invertibility are sufficient for explainability, which is not generally true, specifically because explainability is undefined, complex, and multi-faceted.  I am okay with the current introduction section, but I would still mention somewhere that the two properties you introduced, although reasonable in this context, do not solve the general problem of explainability, just to avoid any further confusion.
> >
> > **SAT & extra saliency approaches.** Thanks, the description in the appendix helps a lot.  I now see that it is inspired by other attribution algorithms.  Table 3 feels complete now that it includes more attribution algorithms and data sets, and seems to show that Bort is not as dependent on the specific choice of SAT.  For these reasons, I will increase my score.
> >
> > **Other changes.** I am happy with all other changes.

---

> > > ### Author Response · Authors · 2022-11-22
> > > **Thanks!**
> > >
> > > We thank the reviewer for the great questions and the patient discussions.
> > > We agree that explainability remains undefined, complex, and multi-faceted.
> > > Our proposed Bort mainly focuses on two topics: parameter comprehensibility (forward, prototype-based) and invertibility (backward, latent state backtracking).
> > > There still remain many research topics about explainability to be studied.
> > > We look forward to clarifying those explainability puzzles in the future!
> > >
> > > Thanks Authors

---

### Official Review · Reviewer_eZA8 · 2022-10-25

**Confidence:** 4
**Correctness:** 4
**Technical Novelty And Significance:** 3
**Empirical Novelty And Significance:** 2
**Recommendation:** 8

**Clarity, Quality, Novelty And Reproducibility:**

Fairly clear, good quality, novel without being ground-breaking, and appears reproducible especially once the code is released.

**Strength And Weaknesses:**

Baking explainability into models by somehow making them inherently interpretable is a relevant problem. The approach proposed in the paper seems sensible and the results look convincing. Some more thoughts:

1. I am not sure if the notions of comprehensibility and transparency are well-established. With that in mind, the presentation of these appears somewhat ad-hoc.

2. While the development appears logical, the end result is quite simple — instead of penalizing the L_p norm of the parameters in the loss function, penalize deviation from orthogonality instead. I wouldn’t be very surprised if a little more algebra reveals some relationship to L_p norm. However, simplicity itself is not something to criticize.

3. While I wasn’t very convinced of the value of the proposal during the development, the results look quite good. Specifically, the improvement in accuracy, improvement in explainability metrics, feature composability, and generation of adversarial examples are all well documented. The only sub-section that seems unsurprising is the verification of properties — orthogonal constraints lead to orthogonality, and the inherent reconstruction constraint leads to good reconstruction.

4. I am not quite sure if I can clearly argue why the technique produces good results. The constraints were developed for one set of objectives yet they somehow improve on another (better established) set as well. A discussion of that and authors’ thoughts on the possible explanation would improve the paper.

5. Minor errors: “possesses comprehensible and transparent simultaneously”, wrong double quotes in “relation”, “only and if only”, “equivalent whether to exert”, “benefits of the two constraints benefit of transparency”, add x-axis label to Fig 3a, “assumption f enough”.

**Summary Of The Paper:**

In this paper, the authors develop notions of comprehensibility and transparency as indicator of explainability, and develop formal notions to introduce these as constraints during model training. The authors also propose a new technique for generating saliency maps. Subsequently, they show that inclusion of these constraints improves accuracy as well as the insertion/deletion metric for quantifying the quality of saliency maps.

**Summary Of The Review:**

Overall, while I think the formalization does reduce to straightforward constraints, the results are (surprisingly) good, and the community could benefit from wider dissemination. I would recommend the paper to be accepted at the venue based on the performance but I would have much appreciated any deeper insights.

---

> ### Author Response · Authors · 2022-11-18
> **Thanks for reviewer eZA8's suggestions and respond to relevant questions**
>
> We thank reviewer eZA8 for the affirmation and pertinent suggestions on our work.
>
> In response to the issues raised by reviewer eZA8, we have made following revisions and additions. Note that we have changed the term "transparency" to "invertibility" following reviewer t9ra's comment. To avoid confusion, we still use the term “transparency” in this supplementary note for Reviewer eZA8.
>
> 1. **Motivations for comprehensibility and transparency**
>     - It should be noted that measuring explainability needs to pass through multiple dimensions. Starting from the two directions (forward and backward) of model calculation features, we focus on two important properties, namely, understandability and transparency.
>     - Comprehensibility is similar to the prototype-based method ([1-2]), except that prototype is a technical description, while comprehensibility is a functional description; at the same time, transparency/invertibility indicates the ability of the model to reconstruct activated signals. For related research, please refer to [3-6].
>     - Therefore, the above two properties are not proposed for the first time in this paper, but we are the first work to define the two properties mathematically for FNN. The advantage of giving a mathematical definition is that it enhances the rigor of the work, and can derive sufficient conditions through the definition so that abstract problems can be solved by algorithms.
>
> 2. **Relationship with $L_p$ norm**
>     - Thanks to the reviewer for the suggestion. We also noticed the relationship with the $L_p$ norm in the derivation. The reason why we choose orthogonality ($L_2$) as the final constraint is mainly for the following two reasons:
>         - **Derivation from sufficient conditions of properties.** According to the definitions of the two properties in this paper (see Definition 1 and Definition 2), we find that the sufficient conditions for the two properties are boundedness and orthogonality constraints. These two constraints can eventually be unified into a single simple constraint. So in the end we chose to use the orthogonality constraint.
>         - **For computational efficiency and simplicity.** We strongly believe that if multiple methods are similar, the easiest method should be chosen. Since our constraints need to be applied to every weight of the model, we need to consider the computational efficiency of the algorithm. The orthogonality constraint of the $L_2$ norm can be calculated efficiently with matrix acceleration by GPUs. Moreover, according to the nature of the normed vector space, $L_p$ norms (when $p>1$) are mutually equivalent norms. So we finally choose to use the orthogonality ($L_2$).
>     - Of course, we admit that there must be some differences between the constraints of $L_p$ ($p\neq 2$) and the constraints of orthogonality ($L_2$), and we think that further research on this difference will be an interesting direction in the future.
>
> 3. **The reason for the performance improvement**
>     - **Explainability improvements.** It is a natural consequence of enhanced comprehensibility and transparency. By aligning the inner product operation to cosine similarity and increasing model reversibility, the inherent explainability of models can be improved. Both quantitative and qualitative experiments confirmed this.
>     - **Performance improvements.** We discover in this work that by choosing appropriate constraints on explainability, the so-called trade-off between explainability and performance can be broken. One intuition is that if constraints are imposed that do not filter out the optimal solution, then the performance remains consistent with the unconstrained model. We present a simple case in Proposition 1, which shows that imposing Bort's constraints on two-layer linear networks does not impair the model's capacity. Moreover, referring to some of the latest research on stochastic configuration networks ([7-8]), it will also be found that the improvement of model performance by orthogonality is universal. But a complete and rigorous theoretical proof of performance improvement is a more difficult theoretical problem to be revealed in future work.
>
> 4. **Mistakes in details**
>     - Thanks to the reviewer for correcting some details of the paper, we have modified them in new version.
>
> [1] Deep learning for case-based reasoning through prototypes: A neural network that explains its predictions
> [2] This looks like that: deep learning for interpretable image recognition
> [3] On pixel-wise explanations for non-linear classifier decisions by layer-wise relevance propagation
> [4] Visualizing and understanding convolutional networks
> [5] Learning important features through propagating activation differences
> [6] Striving for simplicity: The all convolutional net
> [7] Stochastic configuration networks: Fundamentals and algorithms
> [8] Orthogonal Stochastic Configuration Networks with Adaptive Construction Parameter for Data Analytics

---

> ### Author Response · Authors · 2022-11-22
> **Thanks to the reviewer for the helpful review during discussion phase 1**
>
> We thank the reviewer again for the helpful and detailed review.
>
> Since the Phase 1 discussion is over, we just want to ask if the problems raised by the reviewer are properly addressed and if we need to clarify anything else.

---

### Author Response · Authors · 2022-11-18
**Thanks to the reviewers for their meticulous review, we clarify some terms and reiterate the main contribution of the paper**

We thank the reviewers for careful reviews and pertinent suggestions on our work.
In this work, we propose the Bort optimizer to improve model explainability without compromising model performance.
Thanks to reviewer t9ra for the suggestion, we also think that "invertibility" is a more appropriate term than "transparency", so we have substituted the relevant text in the paper.
Since explainability research has many abstract terms, and they often have ambiguous meanings, here we first clarify and explain:

### Clarification of Terms

**Explainability** and **interpretability** are often used interchangeably in many works of literature, although some papers actually point out subtle differences between them.
In this paper, we refer to the definition in [1], where an **interpretation** maps abstract concepts into an understandable domain and an **explanation** reveals the internal mechanism (e.g., how the internal features are calculated by the model).
Our Bort optimizer does not focus on the input/output behavior of the model for mapping the output features back to an understandable format (eg, image and text),
but aims at revealing the internal mechanism of the black-box model by constraining the model parameters.
Specifically, it includes:
1. aligning the inner product operation in FNN to the cosine similarity (comprehensibility);
2. allowing the internal features of the network to recover the features of the previous layer to the greatest extent (transparency/invertibility).

We think that the property pursued by Bort is closer to the "explainability"  in [1] (also similar to "model-centric" in [2]).
Clarifications of related terms have also been added to the supplementary material.

### Our Contributions

It should be emphasized that the description of explainability is a multi-dimensional problem, which is difficult to completely solve with a single indicator.
This work identified two important properties (comprehensibility and transparency/invertibility) from both directions of the model computation process (not to say that these two properties fully constitute explainability).

Our contribution is to mathematically define these two properties aiming at FNNs, so as to obtain sufficient conditions for satisfying these two properties, namely boundedness and orthogonality.
Then, we design an efficient optimizer Bort to impose the boundedness and orthogonality constraints on model parameters.

In this paper, we have conducted both qualitative and quantitative experiments to verify the significant improvement of Bort's method in explainability.
In addition, we demonstrate that through reasonable design, the trade-off between explainability and performance can be broken.
We validate this conclusion by testing classification models on three datasets.

We give an example of two-layer linear networks in Proposition 1 to show why Bort does not limit performance.
By focusing on the research conclusions of stochastic configuration networks ([3-4]), we can also find that orthogonality constraints are indeed consistent in improving model performance.
However, there is still some work to be done to strictly generalize to all nonlinear deep networks.
We think it would be an interesting future direction to investigate a rigorous theoretical analysis of why orthogonality can improve FNNs performance.

[1] Montavon G, Samek W, Müller K R. Methods for interpreting and understanding deep neural networks[J]. Digital signal processing, 2018, 73: 1-15.
[2] Edwards L, Veale M. Slave to the algorithm: Why a right to an explanation is probably not the remedy you are looking for[J]. Duke L. & Tech. Rev., 2017, 16: 18.
[3] Wang D, Li M. Stochastic configuration networks: Fundamentals and algorithms[J]. IEEE transactions on cybernetics, 2017, 47(10): 3466-3479.
[4] Dai W, Ning C, Pei S, et al. Orthogonal Stochastic Configuration Networks with Adaptive Construction Parameter for Data Analytics[J]. arXiv preprint arXiv:2205.13191, 2022.

---

### Decision · Program_Chairs · 2023-01-20

**Decision:**

Accept: poster

**Justification For Why Not Higher Score:**

See above.

**Justification For Why Not Lower Score:**

I've outlined some concerns above; I raised these with the reviewers, and they still favor acceptance. I don't see a reason for me to step in and be an activist AC.

**Metareview: Summary, Strengths And Weaknesses:**

This paper presents a regularizer which tries to make a network's behavior more interpretable by encouraging orthogonality. The reviewers are generally positive about this paper, remarking that it's a simple method that seems to give good results visually and in terms of insertion/deletion metrics. Reviewers did express some concerns about why the technique works, as well as the choice of baselines to compare against. Reviewers had various suggestions of related work, which I encourage the authors to add to the paper. (To this list, I would add "Adversarial robustness as a prior", by Engstrom et al., and past work on orthogonality constraints, such as "Sorting out Lipschitz function approximation," by Anil et al.) The low accuracy of some of the models (for both the proposed method and the baselines) makes me worry a bit about how successful the technique is for modern architectures. But overall, these concerns don't seem critical, and the reviewers generally feel like there is a useful contribution here. I recommend acceptance, and encourage the authors to account for the above points when revising the paper.

**Note From Pc:**

if the above contains the word "oral" or "spotlight" please see: "oral" presentation means -> notable-top-5% and "spotlight" means -> notable-top-25%. As stated in our emails, we are disassociating presentation type from AC recommendations